# Enhanced insights into the genetic architecture of 3D cranial vault shape using pleiotropy-informed GWAS
Seppe Goovaerts [1,2] ✉, Sahin Naqvi [3,4,5,6], Hanne Hoskens [2,7,8], Noah Herrick [9,10], Meng Yuan [1,2,7], Mark D. Shriver [11], John R. Shaffer [10,12], Susan Walsh [9], Seth M. Weinberg [10,12,13], Joanna Wysocka [3,14,15] & Peter Claes [1,2,7,16] ✉

Large-scale GWAS studies have uncovered hundreds of genomic loci linked to facial and brain shape variation, but only tens associated with cranial vault shape, a largely overlooked aspect of the craniofacial complex. Surrounding the neocortex, the cranial vault plays a central role during craniofacial development and understanding its genetics are pivotal for understanding craniofacial conditions. Experimental biology and prior genetic studies have generated a wealth of knowledge that presents opportunities to aid further genetic discovery efforts. Here, we use the conditional FDR method to leverage GWAS data of facial shape, brain shape, and bone mineral density to enhance SNP discovery for cranial vault shape. This approach identified 120 independent genomic loci at 1% FDR, nearly tripling the number discovered through unconditioned analysis and implicating crucial craniofacial transcription factors and signaling pathways. These results significantly advance our genetic understanding of cranial vault shape and craniofacial development more broadly.

The intricacies of human development and evolution are conspicuously evident in the head, where multiple tissue types and organs co-develop and co-evolve in unison to ensure structural and functional coherence[1,2]. For example, the increased brain size in humans, relative to other primates, was accommodated by the co-evolution of craniofacial skeletal features, such as a domed cranium and increased basicranial flexion[2]. Epigenomic divergence related to key craniofacial genes underlies these features and genomic variation related to gene regulation is an important source of craniofacial differences within and between human populations today[3–7]. Knowledge of specific genes and variants and how they affect the different components of the head is key to understanding its development as a complex system.

Formation of the human head starts early in development when the rostral end of the neural tube forms the hindbrain, midbrain, and forebrain, the latter eventually developing into the cerebral hemispheres[8]. From the same neural tube region, cranial neural crest cells (CNCCs) delaminate and migrate ventrally[9,10]. Anterior-most CNCCs form the frontonasal skeleton, while posterior CNCCs populate the pharyngeal arches to form the bone and cartilage of the jaws[10]. The rate of growth of the early brain influences the positioning of facial structures[11], while the flat bones of the neurocranium, derived from the paraxial mesoderm, are joined by flexible sutures that accommodate brain expansion[12]. Dysregulated coordination between the brain and craniofacial mesenchyme results in congenital malformations such as cleft lip and palate or craniosynostosis[11–14]. Throughout

development, many genes have roles that span multiple structures of the human head, influencing both their development and integration. For example, by controlling neural tube development, *ZIC2* and *ZIC3* affect the common origins of the brain and skull[15,16], while *SOX9* has independent roles in brain[17] and facial[18] development. Additionally, a plethora of key facial genes (e.g., *DLX5*, *RUNX2*, and *TWIST1*) and signaling pathways (e.g., BMP/TGF-β, FGF, and Wnt) also affect suture fusion and are implicated in craniosynostosis[18,19]. Together, these extensive genetic and morphological relationships are crucial to consider when studying the genetic basis of craniofacial variation as they present both opportunities and challenges for genetic discovery and interpretation.

Genome-wide association studies (GWAS) have been instrumental in identifying the genetic underpinnings of complex traits[20], including highly heritable cranial vault dimensions[21]. While initial attempts relied on simple anthropometric traits[22–27], a recent GWAS significantly advanced genomic discovery by extracting three-dimensional (3D) cranial vault shape from whole-head magnetic resonance (MR) images[6]. Current findings from GWAS implicate important signaling pathways to contribute to cranial vault shape variation (e.g., FGF, BMP/TGF-β, Wnt, Hedgehog) and affirm the key role of RUNX2, a master regulator of calvarial ossification[28–32]. We have also shown that many of the identified loci are shared between the face, brain, and cranial vault and are closely related to craniosynostosis, in line with current knowledge on developmental biology[6,33]. These findings

suggest that further investigation into cranial vault shape genetics may offer valuable insights into head development as a system and the etiology of craniofacial conditions. However, the tens of loci identified for cranial vault shape remain far fewer than those identified for facial[7,34–43] and brain[33,44–51] phenotypes, revealing a critical gap in cranial vault genetics.

In this study, we aim to leverage prior biological and genetic information to enhance the discovery of genomic loci underlying cranial vault morphology. Given the extensive genomic overlap between brain, facial, and cranial vault morphology[6], as well as evidence that genes associated with bone mineral density (BMD) control cranial suture ossification[52,53], we demonstrate that single nucleotide polymorphisms (SNPs) associated with those traits have an increased likelihood of association with cranial vault shape. In an empirical Bayesian framework, this can be interpreted as evidence in favor of a positive association with cranial vault shape resulting in a posterior, or conditional false discovery rate (FDR) that is decreased with respect to the prior, or unconditioned FDR. We then apply these principles through means of the conditional FDR method[54–61] to a recent cranial vault shape GWAS[6] to leverage GWAS data on brain shape, facial shape, and BMD, thereby revealing novel associated genes and pathways.

## Results

### Cranial vault shape associated SNPs are enriched among SNPs associated with related traits

Using previously described pipelines[6,33,62], we extracted the facial, cranial vault, and mid-cortical surface from the T1-weigthed MR-images of the Adolescent Brain Cognitive Development Study (ABCD Study)[63,64]. Based on the sets of morphological features extracted through principal component analysis (PCA), we estimated the proportion of cranial vault shape variation explained by the brain's shape (Fig. 1a). This estimation was performed cumulatively across the principal components (PCs) of cranial vault shape. Brain shape explained 40.6% of the overall cranial vault shape variation and explained more variation in the first few PCs individually, i.e., up to 61.3% of the variation in the first PC. Furthermore, as shown in Fig. 1b, shape variation explained by the brain at a vertex-level was consistently high across the cranial vault. Together, these findings illustrate that the major modes of cranial vault shape variation, which define its overall dimensions, are reflected in the shape of the underlying cortical surface. In contrast, the 28.1% of cranial vault shape variation explained by facial shape was less focused on the first few PCs (Fig. 1a) and limited mostly to the forehead and regions in close proximity to the face (Fig. 1b).

Next, we investigated whether the genetic architectures of complex traits and diseases were enriched for SNPs associated with cranial vault shape. For this purpose, GWAS summary statistics were obtained from our recent cranial vault shape GWAS[6] and from GWAS studies[7,33,65–73] on other complex phenotypes conducted in independent cohorts. The fold enrichment of statistical association was defined as the proportion of SNPs with $P_{vault} < 0.05$ in the set of SNPs with $P_{other} < 0.05$ versus in the set of all SNPs. We observed a strong enrichment of statistical association with cranial vault shape amongst the SNPs associated with other craniofacial and skeletal traits, including brain shape, facial shape, BMD, and height (Fig. 1c). As these enrichments are directly related to the Bayesian principles of the conditional FDR method used later in this work, we aimed to investigate

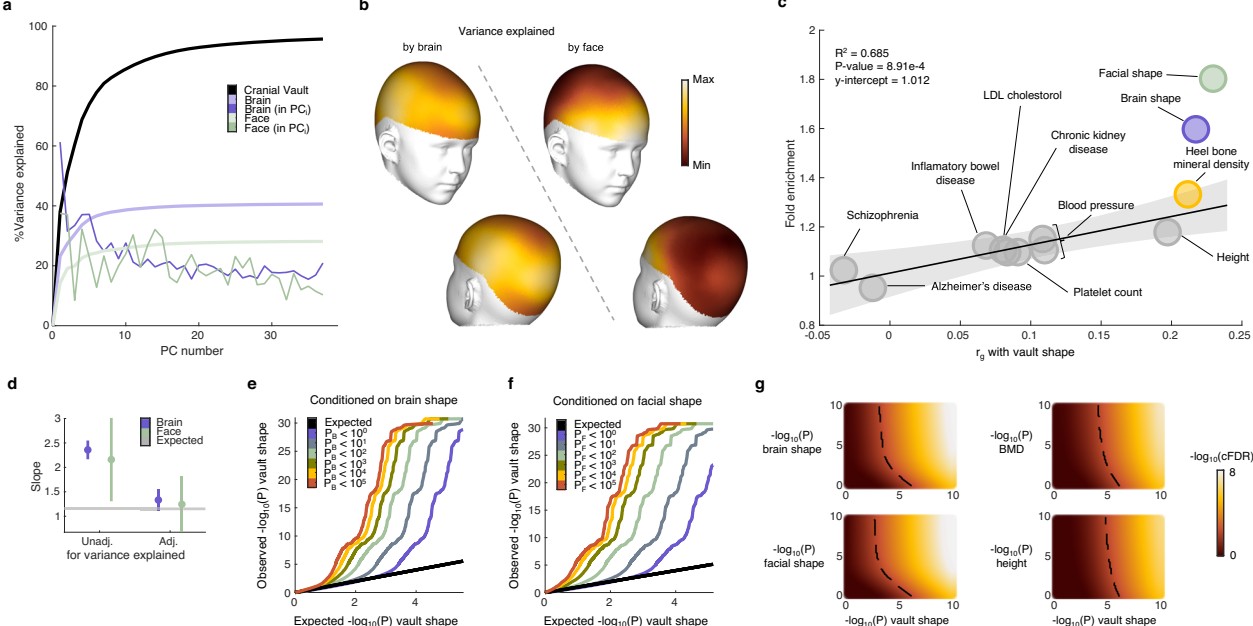

**Fig. 1 | The genetic architectures of brain and facial shape are enriched for SNPs associated with cranial vault shape because of genomic and morphological correlations. a** Cumulative cranial vault (black) shape variation explained by brain (purple) and facial (green) shape. Light green and purple show the total cranial vault shape variation explained by the face and brain, cumulatively over all cranial vault shape PCs. Dark green and purple show the variance explained in the $i^{th}$ PC of the cranial vault. **b** Heatmap of variance explained by brain and facial shape per vertex on the cranial vault. The same color scheme applies to the whole panel. **c** The fold enrichment of SNPs with $P_{vault} < 0.05$ amongst SNPs with $P_{other} < 0.05$ versus all SNPs is plotted against genomic Spearman correlations with cranial vault shape. GWAS sample sizes are as follows: cranial vault shape ($n = 6772$), facial shape ($n = 8246$), brain shape ($n = 19,644$), heel bone mineral density ($n = 426,824$), height ($n = 2,200,007$), systolic blood pressure ($n = 385,798$), diastolic blood pressure ($n = 385,801$), chronic kidney disease ($n = 64,164$ cases + 561,055 controls), platelet count ($n = 408,112$), LDL cholesterol ($n = 361,194$), inflammatory bowel disease ($n = 25,042$ cases + 34,915 controls), Alzheimer's disease ($n = 71,880$ cases + 383,378 controls), schizophrenia ($n = 36,989$ cases + 113,075 controls). The trend line and 95% CI were estimated using iteratively reweighted least squares ($P = 8.91e–4$). **d** GWAS summary data from 63 hierarchical modules of facial shape and 285 hierarchical modules of brain shape were used to estimate and adjust the fold enrichment from (**c**) for the cranial vault shape variance explained by the brain or facial shape modules. Error bars represent the 95% CI. **e** Q-Q plot of $P$ values from the cranial vault shape GWAS conditioned on the $P$ values with brain shape and **f** facial shape. Each color in **e** and **f** corresponds to the set of SNPs from the cranial vault shape GWAS that satisfies the $P$ value criterion in the auxiliary GWAS. **g** Conditional false discovery rate (cFDR) as a function of a SNP's $P$ value for cranial vault shape and for a conditioning trait. Dashed lines indicate the 1% cFDR boundary.

how they corresponded to genomic Spearman correlations[33], another method for assessing genetic overlap that is applicable to multivariate GWAS. Unsurprisingly, genomic Spearman correlations were strongly correlated with cross-trait enrichments of association ($R^2 = 0.685$; $P = 8.91e{-}4$; Fig. 1c). These results demonstrate that when a trait is genetically correlated with cranial vault shape, the information on a SNP's positive association with that trait increases its empirical likelihood to be positively associated with cranial vault shape.

For brain and facial shape, the enrichments were substantially higher relative to their genomic correlations with cranial vault shape. To investigate this overlap further, we downloaded GWAS summary statistics for 63 hierarchical facial shape modules[7] and 285 hierarchical brain shape modules[33], ranging from the entire brain and face to more localized segments, such as the nose, and local regions of the cortical surface (Supplementary Fig. 1). These smaller segments of the brain and face varied in the strength of their genomic and morphological correlations with cranial vault shape, both of which had a positive effect on the enrichments. Moreover, we observed that in the presence of morphological correlations, the observed fold enrichments were inflated relative to what our regression model from Fig. 1c predicted based on the genomic correlation alone. Only after adjusting for the strength of the morphological correlation, did the genomic correlation predict fold enrichments in agreement with our earlier model (Fig. 1d). This illustrates that the high fold enrichments observed for the brain and face could be explained by their morphological correlations with the cranial vault and hence that morphological integration is a significant source of cross-trait associations. We also note that despite the variability in sample size in Fig. 1c, we obtained consistent estimates of the slope when using GWAS data with constant sample sizes (Fig. 1d), demonstrating robustness of the initial estimate.

Next, we visualized the strong enrichment of statistical association with cranial vault shape amongst SNPs associated with brain and facial shape using conditional Q-Q plots[54] (Fig. 1e, f). Under the global null hypothesis, the nominal distribution of P values from any GWAS is expected to follow a uniform distribution, represented by the diagonal line. An increase in tail probabilities due to true genetic associations with cranial vault shape presents itself as a deflection from the expected diagonal line in the Q-Q plots. Here, we observed that the deflection was exacerbated (i.e. shifted leftward) when conditioning on the P values from the brain or facial shape GWAS, i.e., using the P values from those GWASs to define subsets of SNPs from the cranial vault GWAS, indicating an increase in the number of SNPs across the genome showing evidence of association with cranial vault shape. Notably, the magnitude of this leftward shift correlated with the strength of association with brain or facial shape. Previous work by Andreassen et al.[54] demonstrated that such a leftward shift serves as a conservative measure for the FDR of genotype-phenotype associations, with a more pronounced shift indicating a lower FDR.

Building on this, we applied the conditional FDR method[54], leveraging the strength of a SNP's association with auxiliary phenotypes as prior information, or evidence to reduce the prior, or unconditioned FDR of its association with cranial vault shape (Fig. 1g). The heatmaps demonstrate how a SNP's P value (denoted as the minus logarithm) from the cranial vault shape GWAS can be combined with its P value for an auxiliary phenotype to obtain a posterior, or conditional FDR (cFDR). The 1% cFDR boundary in each heatmap, exhibits a saturating behavior, i.e., it approaches a vertical asymptote located at a non-zero minus-log value of the cranial vault P value. This demonstrates that irrespective of the conditional trait, a minimal association in the cranial vault GWAS is necessary for a SNP to attain 1% cFDR. Comparing the 1% cFDR boundaries across auxiliary traits revealed that facial shape and brain shape most effectively reduced this minimal requirement, better than BMD and height (Fig. 1g). Altogether, the results presented in this section clearly demonstrate that the genetic architectures of facial shape, brain shape, and BMD are enriched for associations with cranial vault shape and consequently, that GWAS data on those traits can provide additional evidence for SNPs associated with cranial vault shape.

## Pleiotropy-informed GWAS of cranial vault shape boosts genetic discovery

We conducted a pleiotropy-informed GWAS analysis of cranial vault shape using the conditional FDR method[54], hereafter referred to as cFDR-GWAS, by conditioning a recent cranial vault shape GWAS[6] on one of three auxiliary GWAS datasets: brain shape ("Vault | Brain"), facial shape ("Vault | Face"), or BMD ("Vault | BMD"). Analogous to the previous section, conditioning in this context refers to leveraging auxiliary GWAS data as evidence in an empirical Bayesian framework where the conditional test statistic can be formally defined as the conditional probability (i.e., the Bayesian posterior probability) that a SNP is null for cranial vault shape given that its P values for cranial vault shape and the auxiliary phenotype are equal to or smaller than the observed P values ("Methods"). This probability can be interpreted as the empirical Bayesian generalization of the FDR[74].

As demonstrated above, conditioning on these three traits resulted in the greatest enrichment of statistical association with cranial vault shape. However, there are notable distinctions among these traits. Firstly, while brain shape is not traditionally considered a skeletal trait, it is structurally integrated with the cranial vault and explains a significant portion of its overall shape variance. Secondly, facial shape primarily reflects skeletal features and is predictive for certain aspects of cranial vault morphology (e.g. the forehead region). On the other hand, BMD serves as a systemic skeletal trait which contains limited information about the shape of the cranial vault. Nonetheless, our analysis revealed remarkably similar genome-wide association profiles across all three cFDR-GWAS datasets (Fig. 2), underscoring the robustness of the findings. In total, we identified 120 genomic loci associated with cranial vault shape at a 1% cFDR in at least one of the cFDR-GWAS analyses (Fig. 2a, Supplementary Data 1), a marked increase compared to the 46 loci identified at 1% FDR in the unconditioned GWAS. The expected number of false positive loci at the 1% cFDR cutoff was estimated conservatively as 0.54, ensuring reliable results at this threshold. In total, 90 of the 120 identified loci were not previously mentioned in GWAS of cranial vault morphology. Of these, 75 (83.3%) discoveries can be attributed to the conditional approach, while 15 (16.7%) were simply due to setting the threshold at a 1% cFDR instead of $P < 5e{-}8$. Notably, the cFDR-GWAS analyses that leveraged brain and facial shape data yielded a higher number of genomic loci ($n = 92$ and $n = 83$ respectively) compared to the one using BMD data ($n = 57$) in line with the above results (Fig. 1c). These loci strongly overlapped across all three cFDR-GWAS analyses (Fig. 3a). Furthermore, among the 51 loci identified exclusively in a single cFDR-GWAS at a 1% cFDR threshold, 28 (54.9%) also achieved a cFDR <5% in at least one of the other analyses (Supplementary Data 1).

We investigated the patterns of cranial vault shape variation explained by the lead SNPs identified in both the unconditioned and the cFDR-GWASs of cranial vault shape. Remarkably, these patterns exhibited high concordance across all GWAS analyses, with the greatest variance explained in regions near the Bregma point and the parietal eminence (Fig. 3b). Unsurprisingly, lead SNPs from the cFDR-GWAS based on facial shape excelled in explaining variation across the forehead, while lateral forehead regions remained among the least explained by lead SNPs from other GWASs (Fig. 3b). As a result, the pattern of cranial vault shape variation explained by the cFDR-GWAS based on facial shape was the most dissimilar to that of the unconditioned GWAS, whilst that of the cFDR-GWAS based on BMD was the most similar. In terms of overall shape variation explained by individual lead SNPs of the unconditioned and cFDR-GWAS analyses, similar distributions were observed (Fig. 3c). When combining all the lead SNPs, the unconditioned GWAS explained 2.02% of the overall cranial vault shape variation. This percentage increased to 3.38%, 2.98%, and 2.32% for the lead SNPs from the cFDR-GWASs based on brain shape, facial shape, and BMD respectively, with a cumulative explanation of 4.11% by the combined set of 120 lead SNPs. Despite this substantial overall increase, the patterns of explained shape variation by the cFDR-GWAS lead SNPs remained remarkably similar to the unconditioned GWAS, suggesting that the additional loci exert similar effects on the cranial vault as those identified previously.

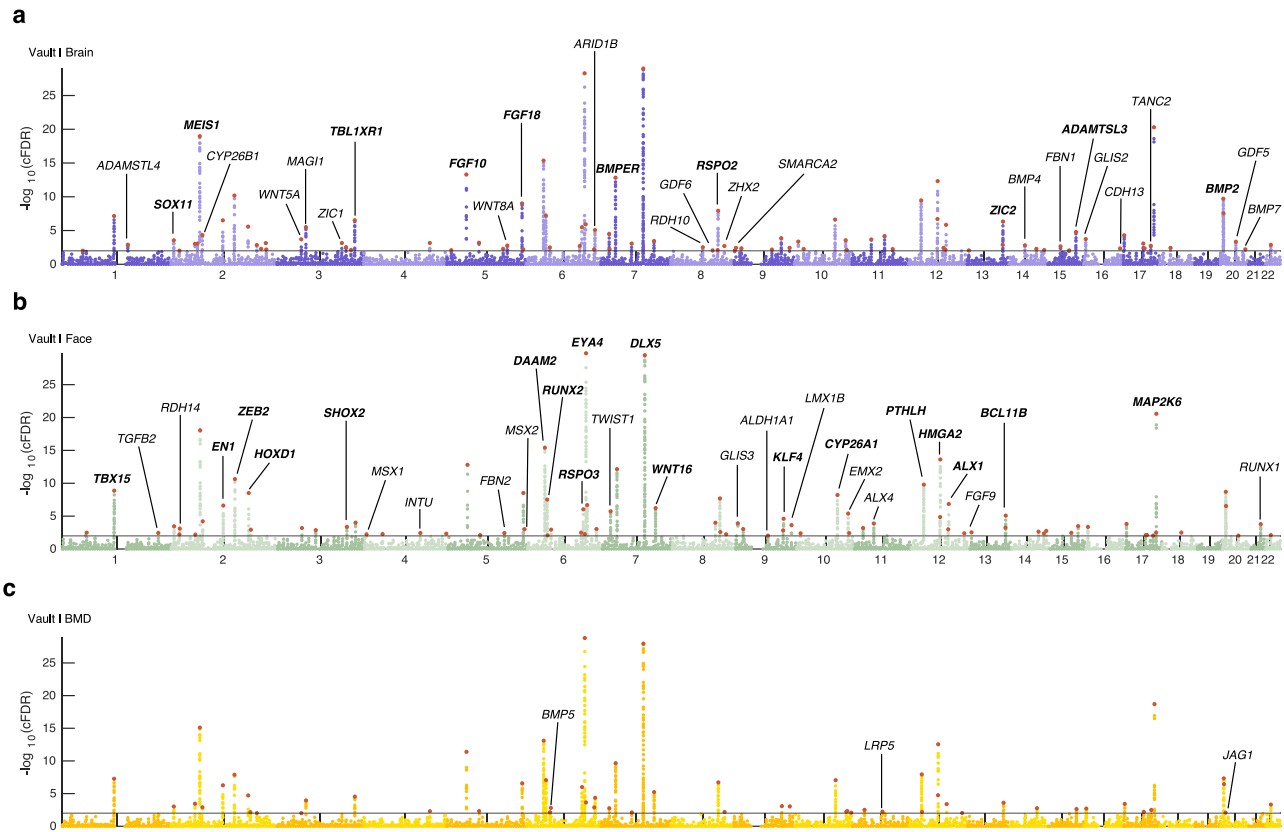

**Fig. 2 | Pleiotropy-informed GWAS of cranial vault shape improves genetic discovery.** Manhattan plots for cFDR-GWAS of cranial vault shape, conditioned on GWAS data on **a** brain shape (purple, "Vault | Brain"), **b** facial shape (green, "Vault | Face"), and **c** heel bone mineral density (yellow, "Vault | BMD"). Horizontal lines indicate the 1% cFDR threshold. Lead SNPs are indicated in red. Candidate genes at a selection of loci are indicated on the Manhattan plot where the lowest cFDR was obtained for that locus. Loci that were previously mentioned in GWAS on cranial vault dimensions are indicated in bold.

We then explored whether the loci identified in each GWAS (at a 1% cFDR) consistently implicated the same biological processes. To accomplish this, we utilized GREAT[75] to annotate Gene Ontology (GO) biological processes to both the unconditioned GWAS and each cFDR-GWAS, tracking the number of overlapping terms (Fig. 3d). Overall, each cFDR-GWAS revealed GO biological processes that strongly overlapped with those identified in the unconditioned GWAS. To further investigate the consistency of the terms identified across each cFDR-GWAS, we aggregated the significant terms (at 5% FDR) into a union set and constructed a single joint semantic space using REVIGO[76]. In this semantic space, Multi-dimensional Scaling (MDS) clustered closely related terms, revealing clusters broadly associated with development, signaling, and regulation (Fig. 3e). In this space, we then indicated the top 100 terms from each cFDR-GWAS separately. While it should be expected that each cFDR-GWAS yields a slightly different set of processes resulting from the identification of unique loci, Fig. 3e demonstrates that these unique processes remain closely related to those identified by the other cFDR-GWAS. In essence, while conditioning on different auxiliary phenotypes led to partially overlapping sets of loci, the choice of secondary phenotype did not result in divergent biological findings.

**Pleiotropy-informed GWAS implicates key signaling pathways in craniofacial development**

The use of FDR-based statistics in GWAS allows for straightforward control of the number of falsely identified loci. While it is common practice in GWAS to use a very strict significance threshold to declare positive associations, we reasoned that relaxing the threshold to 5% cFDR (instead of 1% cFDR) would yield a larger set of genomic loci, still enriched for biologically meaningful information[77], that could help to more robustly implicate

biological processes and pathways. Therefore, to further explore the SNPs and genes underlying cranial vault shape, we lowered the significance threshold to 5% cFDR in each cFDR-GWAS and subsequently merged their genomic loci into 328 independent loci (Supplementary Data 1), explaining 8.53% of overall cranial vault shape. Following an enrichment analysis using GREAT[75], we observed, indeed, that most of the top 20 GO biological processes and the top 20 craniofacial mouse phenotypes identified for the initial set of 120 loci were consistently and more strongly supported among the broader set of 328 loci (Fig. 4a, b; Supplementary Data 2–5). Furthermore, among the loci identified at a 5% cFDR, we identified additional genes linked to craniosynostosis in humans, including *PPP3CA*[78], *CDKN1C*[79], *FOXP1*[80], *PRRX1*[80], and *ZBTB20*[80]. Taken together, these results show that relaxing the significance threshold to 5% cFDR helped strengthen the support for key biological processes and pathways through the identification of additional, biologically meaningful candidate genes.

Among the genes near the identified loci were many members of the Wnt, BMP/TGF-β, and FGF signaling pathways as well as several genes related to metabolism of retinoic acid (Fig. 4c), which itself is a key signaling molecule during craniofacial development. We found a wide variety of genes involved in both beta-catenin dependent and independent Wnt signaling including ligands (*WNT3A*, *WNT5A*, *WNT8A*, *WNT9A*, *WNT16*, *RSPO2*, and *RSPO3*), antagonists (*SOST* and *SFRP2*), (co-)receptors (*FZD4*, *FZD7*, and *LRP5*), and other genes related to signal transduction (*PRICKLE2*, *DAAM1*, *DAAM2*, *CCND1*, and *PPP3CA*). Furthermore, we identified several *fibroblast growth factors* (FGFs) with known roles during craniofacial development, including *FGF9*, *FGF10*, *FGF12*, and *FGF18*. Notably, none of their receptors were identified despite their prominent roles in cranial vault development as evidenced by their causative roles in craniosynostosis[81]. From the BMP/TGF-β signaling pathway, we identified several extracellular

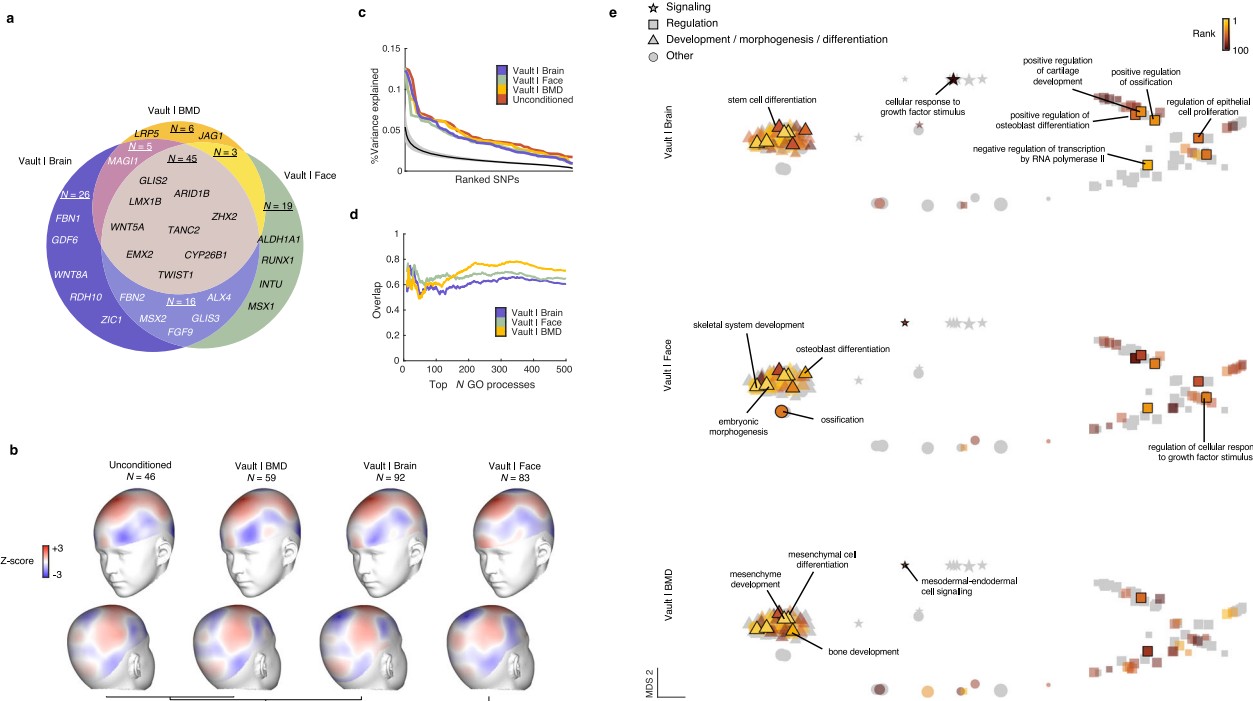

**Fig. 3 | Pleiotropy-informed GWAS of cranial vault shape yields consistent results across auxiliary traits. a** Venn diagram showing locus overlap between the cFDR-GWAS analyses in Fig. 2. Venn diagram dimensions are proportional to the number of loci. A selection of genes not previously mentioned in cranial vault GWAS literature is shown. **b** Patterns of variance explained by the set of lead SNPs from each cFDR-GWAS. The vertex-wise variance explained was transformed into Z-scores for each cFDR-GWAS independently. Towards red indicates high explained variation by the lead SNPs of each respective GWAS. Vertex-wise patterns of Z-scores were clustered based on pairwise correlations. **c** Distributions of the variance explained by individual lead SNPs of each GWAS. Solid black line and gray band give the mean and 95% confidence interval for the null estimate through 1000 random genotype permutations. **d** Proportional overlap between the sets of the top N most

enriched GO biological processes obtained for the cFDR-GWASs and that obtained for the unconditioned GWAS. **e** Semantic space of the joint set of GO biological processes that were significantly enriched (at 5% FDR) amongst the loci from each individual cFDR-GWAS in Fig. 2. The space was obtained using REVIGO based on multidimensional scaling (MDS) of the pairwise similarities between GO biological processes in the set. Different symbols indicate different broad categories of processes, and symbol size corresponds to the total number of genes for each process. Corresponding to each cFDR-GWAS, the top 100 most enriched processes are indicated and colored by their rank. A selection of processes is annotated in the plot where the process attained the lowest rank. Lead SNPs for the "unconditioned" GWAS in (**b–d**) were called at 1% FDR to provide a fair comparison.

signaling ligands, including *growth differentiation factors* (GDFs; *GDF5* and *GDF6*), *bone morphogenic proteins* (BMPs; *BMP2*, *BMP4*, *BMP5*, and *BMP7*), and *TGFB2*, as well as one BMP-receptor (*BMPR2*), signaling inhibitors (*NOG* and *BAMBI*), and modulators of BMP/TGF-β signaling (*BMPER*, *FBN1*, *FBN2*, and *THSD4*). Lastly, we identified several genes related to the biosynthesis (*ALDH1A1*, *RDH10*, and *RDH14*) and degradation (*CYP2A1*, and *CYP26B1*) of retinoic acid.

### Cross-trait associations are specifically enriched at mesenchymal TF targets

Our previous GWAS on cranial vault shape identified a locus near *RUNX2*, encoding an osteogenic transcription factor (TF) which is expressed in the craniofacial mesenchyme, but not in the brain. Nonetheless, we found that this locus overlaps with brain shape[33] and facial shape (most strongly with the nose)[7] (Fig. 5a). The lead SNPs from the cranial vault shape (rs3799970) and brain shape (rs542444) GWASs are in near perfect linkage (D' = 0.994; r² = 0.792; 1000 G EUR populations), affirming that this is the same locus. The facial shape lead SNP, rs227832, was located ~130 kb away from rs3799970, but still in strong linkage (D' = 0.889; r² = 0.500; 1000 G EUR populations). Next, we examined if the binding sites of RUNX2 were specifically enriched for cross-trait associations between the cranial vault and either the brain or face, similarly to Fig. 1c. To this end, we obtained binding site coordinates for RUNX2 and 194 other TFs from TFLink[82] and calculated the cross-phenotype enrichments across the binding sites of each TF. Brain-associated and face-associated SNPs at RUNX2 binding sites were significantly more enriched for vault-associated SNPs (Fig. 5b and

Supplementary Data 6;1.98-fold and 2.77-fold respectively) compared to those at other TF's binding sites ($P_{brain}$ = 1.75e−4; $P_{face}$ = 2.15e−7; one-tailed *t* test). This suggests that activity of the osteogenic TF, RUNX2, underlies in part the cross-trait enrichment of associations.

Despite its well-known role in craniofacial skeletal development and its causative role in syndromic craniosynostosis, TWIST1 was not implicated by any GWAS of cranial vault dimensions previously. However, by following a pleiotropy-informed GWAS approach, we did identify a locus near *TWIST1*. Using stratified linkage disequilibrium score regression (S-LDSC) on the summary statistics from our previous cranial vault shape GWAS[6], we examined if genomic targets regulated by TWIST1 were disproportionately enriched for heritability. From a previous publication[83], we obtained the coordinates of distal assay for transposase accessible chromatin (ATAC) peaks that were differentially accessible upon loss or acute depletion of TWIST1 in human CNCCs. Figure 5c shows how the down-regulated TWIST1-dependent peaks were specifically enriched for cranial vault shape heritability (TWIST1 loss: 20.7-fold, *P* = 6.87e−4; and TWIST1 depletion: 20.3-fold; *P* = 1.25e−3, one-tailed *t* test). It was previously shown[83] that the down-regulated, but not the upregulated peaks were highly enriched for TWIST1 binding motifs. Therefore, these findings provide orthogonal validation that regulation by TWIST1 plays a role in typical-range cranial vault shape variation.

Furthermore, SNPs at the downregulated TWIST1 peaks were also more enriched for associations with cranial vault shape compared to the genome-wide estimate (Fig. 5d), both in the brain shape GWAS (TWIST1 loss: 1.68-fold, *P* = 0.0495; TWIST1 depletion: 1.71-fold, *P* = 8.75e−3; one-

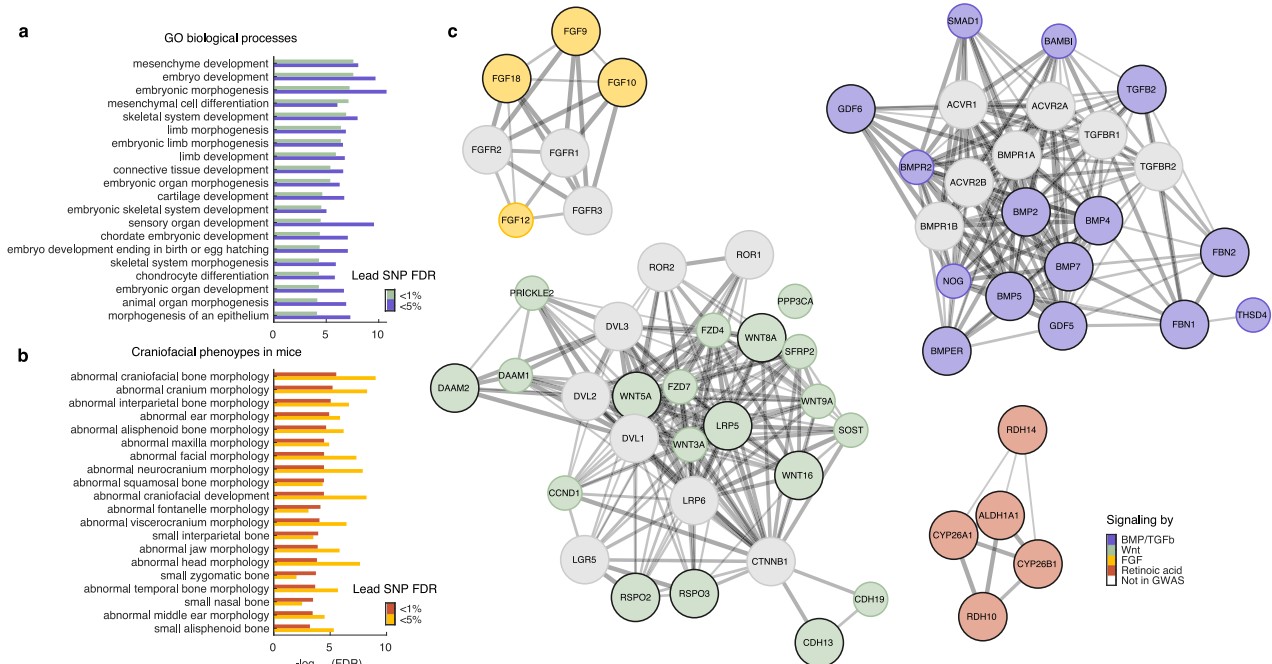

**Fig. 4 | Pleiotropy-informed GWAS implicates major signaling pathways involved in craniofacial development. a** Enrichment of GO biological processes amongst cFDR-GWAS loci identified at 1% and 5% FDR. Only the top 20 processes are shown. **b** Enrichment of mouse phenotypes amongst cFDR-GWAS loci identified at 1% and 5% FDR. Only the top 20 craniofacial phenotypes, selected from the full list of phenotypes, are shown. **c** STRING networks representing members of the

BMP/TGF-β, Wnt, Retinoic acid, and FGF signaling pathways identified in the cFDR-GWAS. Genes in large, colored nodes with a black border were identified at 1% cFDR; those in smaller, colored nodes without a black border were identified at 5% cFDR; and genes in white nodes were not identified in the GWAS but are added to provide context.

tailed *t* test) and facial shape GWAS (TWIST1 depletion: 1.91-fold, $P = 0.0387$; one-tailed *t* test). Even after Bonferroni-correction, the 1.71-fold enrichment of vault-associated SNPs among the brain-associated SNPs at down-regulated peaks upon acute TWIST1 depletion was still significantly higher than the genome-wide estimate of 1.59 ($P_{adj} = 0.0350$; one-tailed *t* test). Together, these analyses demonstrate that SNPs located at the binding sites of mesenchyme-specific TFs contribute strongly to the cross-trait genetic enrichment between the cranial vault and both the face and brain. These findings align with the strong enrichments of mesenchyme-related processes among the genes located near the cFDR-GWAS loci.

## Discussion

In this work, we utilized a pleiotropy-informed approach to enhance the detection of SNPs associated with cranial vault shape and to deepen our genetic understanding of this understudied component of the craniofacial system. The conditional FDR method, inspired by the empirical Bayes framework, states that the FDR of genotype-phenotype associations may be reduced by incorporating prior information on the SNPs as evidence, specifically regarding whether they have prior associations with closely related traits. Here, we first demonstrated that SNPs associated with traits closely related to cranial vault shape are indeed more likely to be associated with it. We then conditioned a recent cranial vault shape GWAS[6] on the *P* values from a facial shape[7], brain shape[33], and bone mineral density[65] GWAS, yielding 120 independent loci at a 1% cFDR, and 328 at a 5% cFDR, substantially more than previously identified[6,22–27].

Conditioning on GWAS data from three distinct phenotypes was proven valuable not only to improve genomic discovery, but also in demonstrating the robustness of our findings. Specifically, our analyses revealed remarkably consistent genome-wide association profiles, biological processes, and phenotypic effects across all cFDR-GWASs. This consistency underscores the robustness of the genetic associations identified and suggests that the observed associations between genetic variants and cranial vault shape are not solely influenced by the specific traits used for

conditioning. In fact, these findings illustrate that cross-phenotype genetic associations are abundant among different constituents of the craniofacial system and bone-related measurements more broadly and that they could effectively be leveraged to boost GWAS discovery. This aligns with the widespread pleiotropy of craniofacial transcription factors, genes involved in system-wide skeletal development, and the mutual influences between the face, brain, and cranial vault that occur on structural and physiological levels during head development[1,11,33]. Still, we note a few examples where the choice of auxiliary trait was directly related to the findings. Only when leveraging BMD, we identified a locus near *LRP5* at a 1% cFDR, for which loss-of-function mutations are associated with osteoporosis and gain-of-function mutations with higher bone mass and craniosynostosis[84,85]. Additionally, a locus near *MSX1* was only identified at a 1% cFDR by leveraging facial shape. This gene plays a key role in facial development, including the frontal bone[86], with mutations linked to multiple craniofacial anomalies such as cleft lip and palate[87].

Only after the forebrain initially forms from the rostral end of the neural tube do cranial neural crest cells and mesodermal progenitor cells give rise to much of the craniofacial skeleton[8,10,88]. From that point onward, neurological and mesenchymal tissues develop with tight coordination[1,11]. Here, we have demonstrated how a substantial proportion of cranial vault shape information is embedded in the brain's shape as a product of this co-development and how the regulatory landscapes of mesenchymal TFs are enriched for their cross-trait genetic associations. Together, these findings suggest that brain shape, in part, reflects the development of the craniofacial skeleton and consequently, that brain-derived traits subjected to GWAS may implicate cranial vault-associated loci. In fact, examples of such loci have been reported previously[6,33], including *RUNX2*, *TWIST1*, and *ALX1*, which are not expressed in the brain but play prominent roles in craniofacial skeletal development[28,83]. More generally, several brain shape GWAS studies have found skeletal processes to be overrepresented among the identified genes[33,44]. These findings could arise from mediated pleiotropy[89], where a typically well-powered brain GWAS picks up on the skeletogenic genes that

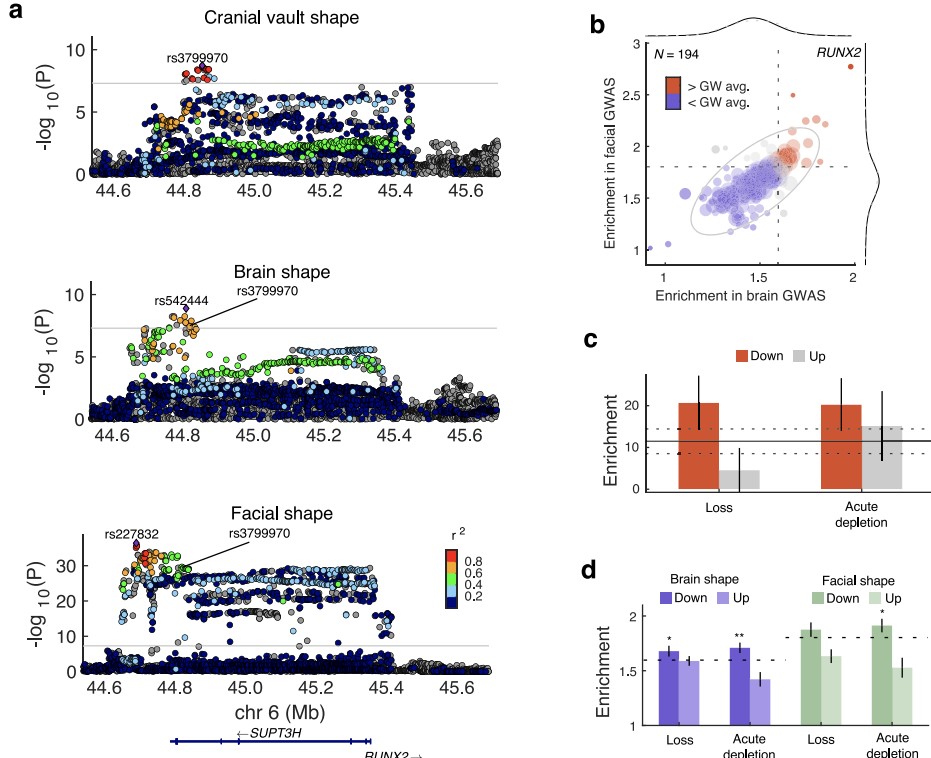

**Fig. 5 | Cross-trait associations with cranial vault shape are specifically enriched at binding sites of mesenchymal transcription factors. a** LocusZoom plots show locus around rs3799970, near RUNX2 in the cranial vault, brain, and facial shape GWAS. Colors represent the LD ($r^2$; 1000G EUR populations) with the lead SNP from each GWAS. SNPs in gray did not overlap with 1000G reference set. The horizontal line indicates the $P < 5e-8$ threshold. **b** Biplot showing fold-enrichment of statistical associations between cranial vault shape and brain or facial shape, but specifically for the SNPs located in the binding domain of various TFs ($N = 194 + RUNX2$). Dashed lines indicate the genome-wide estimate which is the same as in Fig. 1c. Empirical distributions (gray dashed lines) on top of fitted t-distributions (black full lines) are show for both axes. The ellipse indicates the empirical 95% confidence boundary. **c** Fold enrichment of cranial vault shape heritability (from Goovaerts et al.[6]) in distal ATAC peaks differentially accessible upon TWIST1 depletion ($N_{up} = 15,727$; $N_{down} = 35,553$) or loss ($N_{up} = 38,236$; $N_{down} = 31,689$). Error bars indicate the standard error. The horizontal line indicates the enrichment in all human CNCC distal ATAC peaks ($N = 189,601$), with flanking dashed lines the standard error. **d** Fold-enrichment of statistical associations between cranial vault shape and brain (purple) or facial shape (green), but specifically for the SNPs located at distal ATAC peaks differentially accessible upon TWIST1 depletion or loss. Error bars indicate standard errors estimated over 1000 bootstrapped sets of peaks. Dashed horizontal lines indicate the genome-wide estimate which is the same as in Fig. 1c. (*$P < 0.05$, **$P < 0.01$, one-tailed $t$ test).

affect brain shape through their effects on the cranium. Here, it is likely that mediated pleiotropy underlies, at least partially, the many cross-phenotype genetic associations between the brain and cranial vault and therefore the success of the conditional FDR method.

Both the vicerocranium and the vault portion of the neurocranium form through intramembranous ossification, i.e., the direct ossification of mesenchymal cells[88]. This process is regulated by osteogenic transcription factors, such as RUNX2 and SP7, as well as extracellular signaling by FGF, BMP, Wnt, and IHH[29,90]. Additionally, many of these genes affect bone mineral density across the body through processes including bone formation, homeostasis, and remodeling[91,92]. Moreover, craniofacial transcription factors from the *basic helix-loop-helix* and *homeodomain* families cooperatively control the development of neural crest-derived mesenchyme in the face and cranial vault[83,93]. This high degree of biological pleiotropy underlies the genomic overlap between these skeletal phenotypes and therefore contributes to the effectiveness of the conditional FDR method. Additionally, since the definitions of the cranial vault and face in the original GWASs partially overlap at the forehead, any locus affecting that area would automatically be associated with both phenotypes but would not always be detected if statistical power is low. This likely constitutes a secondary reason why leveraging facial shape was effective in a cFDR-GWAS.

Our pleiotropy-informed GWAS identified a plethora of loci near genes with well-known roles in cranial vault development. Most notably, we identified a locus near *SP7* and two loci near *RUNX2*, which encode transcription factors essential for osteoblast differentiation and bone formation[94]. In addition, many of the other genes identified have direct or indirect effects on the activity of *SP7* and *RUNX2*. For example, signaling by BMPs and IHH are known to induce expression of *SP7* in a manner dependent on RUNX2, or independent of RUNX2 through MSX2[95–99]. Meanwhile, reciprocal induction of *RUNX2* expression and signaling by Wnt, FGF, and PTHLH allow for precise control of osteogenesis[30,31,90]. Mutations in several of these genes have been linked to craniosynostosis, i.e., the premature fusion of one or more cranial sutures, a condition that affects approximately 1/2500 births[100]. Interestingly, genes that likely induce *RUNX2* or *SP7* expression tend to cause craniosynostosis through gain-of-function mutations or whole gene duplications, such as *MSX2*[101,102], *LRP5*[85], *SOX11*[103], *IHH*[104], and *RUNX2* itself[105,106]. The opposite phenotype is observed for deletions in *MSX2*[107] or *RUNX2*[108], suggesting a dosage-sensitive effect of both genes on cranial suture fusion[81]. Along the same lines, loss-of-function mutations in *TWIST1*, which directly antagonizes RUNX2 in the developing coronal suture, cause Saethre-Chotzen syndrome, characterized by coronal craniosynostosis[109,110]. Given that the genes near many of the loci identified in our pleiotropy-informed GWAS are directly or indirectly related to *RUNX2* or *SP7*, it is possible that together they can predict a considerable proportion of craniosynostosis risk. In fact, GWASs on non-syndromic craniosynostosis have identified risk SNPs near *BMP2*, *BMP7*, *BMPER*, and *DLX5*, which overlap with loci identified here, and are directly linked to *RUNX2*[6,111–115]. Due to their large effect sizes, these loci could be identified in relatively small

patient cohorts, however, it is likely that many more loci are involved in non-syndromic craniosynostosis risk, such as other loci identified here. Furthermore, recent reports have noted that normocephalic sagittal craniosynostosis is underdiagnosed and therefore more prevalent than previously thought, with estimates as high as 4.7% in the overall population[116,117]. Consequently, our pleiotropy-informed GWAS may pick up on craniosynostosis-related signals directly. In addition to contributing risk, the polygenic background comprised of the identified loci may be predictive for phenotype severity, an idea already demonstrated in mice[118].

In addition, we identified genes involved in the two-step oxidation of retinol (vitamin A) into the bioactive retinoic acid. Specifically, *RDH10* and *RDH14* encode enzymes that oxidize retinol into retinaldehyde[119,120], which is further oxidized into retinoic acid by ALDH1A1[119]. Previous studies have found that retinoic acid interacts with the Wnt, Hedgehog, and BMP pathway during osteogenesis[121,122], whereby excess retinoic acid causes skeletal anomalies that are phenocopied by loss-of-function mutations in *CYP26A1* and *CYP26B1*, which encode enzymes involved in retinoic acid degradation[119,123,124]. Notably, retinoid-induced craniosynostosis has been observed in the case of homozygous mutations in *CYP26B1*[124–126]. These findings suggest that genes involved in regulating retinoic acid levels during embryonic development contribute not only to craniofacial anomalies, but also to the variation in cranial vault shape seen in the general population. Since maternal diet affects embryonic retinoid availability[127], retinoid-related effects on cranial vault shape are likely, to some extent, mediated by gene-environment interactions.

As with other GWAS studies, replication of the genetic findings is key to confirm their validity. Unfortunately, the limited availability of linked craniofacial and genetic data poses significant challenges for both genetic discovery and replication efforts, especially for the cranial vault. While 3D facial data can be collected through 3D surface scanning, this is much more difficult to achieve for the cranial vault due to the presence of hair. Alternatively, medical imaging provides a 3D view of the vault, but has other limitations, e.g., computed tomography (CT) imaging exposes participants to radiation and MR-imaging is expensive. A workaround is to use existing brain MR-image collections to extract the cranial vault[6], a task that poses technical challenges since imaging protocols are optimized regarding the brain. Moreover, due to privacy concerns, these images are often anonymized by removing parts of the face and ears, regions that overlap with the cranial vault[128]. While additional data sources and larger sample sizes are necessary to further elucidate the genetic architecture of complex traits in general, multi-trait GWAS strategies already enhance genetic insights in the meantime[54,129–131]. In this work, we specifically opted for the conditional FDR method as it could seamlessly deal with the multivariate GWAS statistics of cranial vault shape and the other morphological phenotypes. We opted for a threshold of 1% cFDR to call statistical significance of individual loci, whereby fewer than one locus among the 120 independent loci is expected to be a false positive. At 5% cFDR, the expected number of false discoveries was still low, hence, we used these loci to implicate additional candidate genes related to the already identified pathways. This level of confidence may be too low to warrant the time and cost-intensive functional follow-up of individual loci. However, collectively, our loci may form the basis for candidate gene studies or polygenic risk scores (PRS) in small cohorts of patients with craniofacial conditions, e.g., craniosynostosis, where they could directly lead to enhanced etiological understanding and risk prediction.

In summary, the conditional FDR method allowed the use of prior knowledge to inform on genetic associations with human cranial vault shape, resulting in an enhanced discovery of SNPs. The identified genes robustly implicate the Wnt, BMP/TGF-β, FGF, and retinoic acid pathways to play key roles in shaping the vault. Given the close link between many of the identified loci and the master regulators of suture ossification, *RUNX2* and *SP7*, it is plausible that collectively our loci comprise a polygenic background predictive of craniosynostosis risk and severity. In sum, our improved GWAS discovery substantially enhances the genetic understanding of typical-range variation in cranial vault shape and craniofacial development more broadly.

## Methods

### Ethics statement
Data available through the controlled access NIMH data archive has been approved for broad sharing. Local institutional approval (S60568) was granted for access to this data. Legal guardians of ABCD study (https://abcdstudy.org/about/) participants provided written informed consent to participate and have their data shared.

### Cohort data
For investigations into the proportion of explainable vault shape variation by genetic and phenotypic variables, we used participant data from the ABCD Study, a 10-year longitudinal study following brain health and development through adolescence[63]. A variety of data from 11,880 nine- and ten-year-old boys and girls has been collected across 21 sites in the US and have been made available through the controlled-access NIMH data archive (https://nda.nih.gov/). The data used in this work, including T1-weighted MRIs, genotypes, and additional information on sex at birth, age, weight, height, etc. were obtained from data release 3.0 (February 2020).

### GWAS summary data
GWAS summary statistics were obtained from our recent cranial vault shape GWAS[6] and from recent, large-scale GWAS studies[7,33,65–73] on other complex phenotypes conducted in independent cohorts of predominantly recent European ancestry. This includes the cranial vault GWAS in the ABCD cohort, for which ~80% of the alleles were derived from recent European ancestors[6], which forms the basis for the main analyses performed in this work. Additionally, a version of this GWAS, conducted in a sub-sample with inferred European ancestry, was used only for S-LDSC. All GWAS data used in this work are freely available online. An overview of studies and sample sizes is given in Supplementary Table 1.

### Image processing and phenotyping
From the ABCD study's T1-weighted MR images, we extracted the mid-cortical surface using FreeSurfer v6.0.0[132], as well as the facial and cranial vault surface using our previously described pipeline[6,62] and Meshmonk[133]. Only participants with inferred recent European ancestry were retained as abundant imaging artifacts can disproportionately affect individuals of different ancestries due to differences in hair types and facial structures, increasing the risk of spurious results. A detailed description of image processing and sample selection can be found in the Supplementary Note and Supplementary Fig. 2.

Quality controlled 3D surface meshes of the brain ($n = 29,759$ vertices), face ($n = 7160$ vertices), and cranial vault ($n = 11,410$ vertices) were Procrustes superimposed separately and then bilaterally symmetrized. After removing a set of related individuals up to the third degree using KING[134] (cutoff = 0.0442), the resulting surfaces were adjusted for covariates, including sex at birth, age, weight, height, site/MRI machine, brain/vault/facial size, and genomic ancestry using partial least squares regression (Matlab 2023a, *plsregress*). Due to the abundance of facial artifacts, facial surfaces were additionally adjusted for artifact-related variables (Supplementary Note and Supplementary Fig. 2) and their interaction terms with BMI, head size and facial size. Finally, we extracted the optimal number of variables to describe the phenotypes using PCA and parallel analysis on all three phenotypes separately.

### Cross-phenotype explained variance
Using partial least squares regression (Matlab 2023a, *plsregress*) and data from 1969 individuals with high quality brain, facial, and cranial vault data available, we estimated the percentage of cranial vault shape variation explained in each principal component by the full set of principal components from either the brain or face. The final PCA models contained 37, 46, and 286 principal components for the vault, face, and brain, explaining 95.7%, 96.7%, and 77.1% of their morphological variation respectively.

## Genomic Spearman correlations

Unlike most GWASs on univariate traits, the multivariate GWAS on cranial vault shape did not yield signed effect sizes. Therefore, the commonly used approach of calculating genomic correlation using linkage disequilibrium (LD) score regression[135] was not applicable here and instead we opted to calculate genomic Spearman correlations as described by Naqvi et al.[33]. Briefly, SNPs were intersected with the HapMap3[136] set of SNPs excluding any SNP within the major histocompatibility region (GRCh37 positions of chr6:25,119,106–33,854,733). The remaining SNPs were then organized into 1725 approximately independent LD blocks[137] estimated in European populations[138]. We used European-derived LD blocks since all GWAS data was generated in samples of predominant recent European ancestry.

The genomic Spearman correlation (Matlab 2023a, *corr* with parameter '*Spearman*') between both traits and using $n$ LD blocks was calculated based on Eq. 1, where $d_i \equiv R[-\log_{10}(P_{i,x})] - R[-\log_{10}(P_{i,y})]$, the difference in rank between the average $-\log_{10}(P)$-value of traits $x$ and $y$ in LD block $i$.

$$r_g(x,y) = 1 - \frac{6 \sum_{i=1}^{n} d_i}{n(n^2 - 1)} \tag{1}$$

## Cross-trait enrichment of statistical association

The enrichment of statistical association with cranial vault shape among the SNPs associated with another trait was defined as the fold increase in the proportion of SNPs associated with cranial vault shape (at $P_{vault} < 0.05$) among the SNPs associated with an auxiliary trait (at $P_{other} < 0.05$) relative to the set of all SNPs (Eq. (2)). The resulting fold-enrichment essentially measures the increase in tail-probabilities in the $P$ value distributions of the subset of SNPs relative to the full set. By Bayes' theorem, it follows that this definition of the cross-trait fold-enrichment is reciprocal (Eq. (2)), however, this property is not further utilized in this work.

$$fold\ enrichment = \frac{Prob(P_{vault}<0.05\,|P_{other}<0.05)}{Prob(P_{vault}<0.05)} = \frac{Prob(P_{other}<0.05\,|P_{vault}<0.05)}{Prob(P_{other}<0.05)} \tag{2}$$

Similarly, the cross-trait enrichment specific to a TF, $t$'s binding sites, $B_t$, was obtained by conditioning the probabilities in Eq. (2) on the prerequisite that a SNP, $s$, is in $B_t$ (Eq. (3)).

$$fold\ enrichment = \frac{Prob(P_{vault}<0.05|P_{other}<0.05,\ s \in B_t)}{Prob(P_{vault}<0.05\,|\,s \in B_t)} \tag{3}$$

Binding sites for various TFs ($N = 194 + \text{RUNX2}$, after excluding TFs with <500 binding sets across the autosome) were obtained from TFlink[82] v1.0 (https://cdn.netbiol.org/tflink/download_files/TFLink_Homo_sapiens_bindingSites_All_annotation_v1.0.tsv.gz). The binding sites of RUNX2 were obtained through a chromatin immunoprecipitation (ChIP) assay in osteosarcoma cells[139]. Distal hCNCC ATAC peaks differentially accessible upon TWIST1 loss or acute depletion were obtained from Kim et al.[83]. In both cases, we selected all SNPs within a margin of 10 kb around each binding site or peak.

## Conditional FDR analysis

As originally introduced by Andreassen[54], the conditional FDR of a genotype-phenotype association is defined as the conditional probability (i.e., the Bayesian posterior probability) that a SNP is null for the target phenotype given that the $P$ values for the target and auxiliary phenotype are equal to or smaller than the observed $P$ values. Formally, using the cranial vault as the target phenotype, this can be written as

$$FDR(P_{vault}|P_{other}) = \frac{\pi_0(P_{other})F_0(P_{vault}|P_{other})}{F(P_{vault}|P_{other})} \tag{4}$$

where $F_0$ and $F$ are the conditional cumulative density functions (CDF) for null and all SNPs respectively, and $\pi_0(P_{other})$ is the proportion of SNPs that are null for the cranial vault given that the $P$ value for the auxiliary phenotype is equal to or smaller than $P_{other}$. Assuming independence, $F_0(P_{vault},\,|,P_{other})$ simplifies to $P_{vault}$. Moreover, to estimate $FDR(P_{vault}|P_{other})$ conservatively, $\pi_0(P_{other})$ is set to 1 and $F(P_{vault}|P_{other})$ is replaced with the empirical conditional CDF. This formulation is a generalization of the empirical Bayesian interpretation of the FDR by Efron[74].

The conditional FDR analysis was conducted based on the *pleioFDR*[54] software (https://github.com/precimed/pleiofdr). Due to complex Linkage Disequilibrium (LD) that can potentially bias FDR estimates, SNPs within the major histocompatibility region on chromosome 6 and chromosome 8p23.1 (GRCh37 positions of chr6:25,119,106–33,854,733 and chr8:7,200,000–12,500,000, respectively) were removed before fitting the conditional FDR models. The final models were constructed based on 500 iterations of randomly LD-pruned ($r^2 < 0.2$) SNPs.

## Identification of independent genomic loci

Independent loci for each cFDR-GWAS were identified using FUMA[140] v1.6.1 with default settings. In summary, independent lead SNPs were first identified based on LD estimated within the European samples from the 1000 Genomes Project Phase 3[138] dataset using a cutoff of $r^2 = 0.1$. Next, independent loci were obtained by merging any lead SNPs within 250 kb into the same locus represented by the most significant lead SNP. Any reference to "lead SNP" in the rest of the manuscript refers to only the most significant lead SNP at each independent locus. The final set of independent loci across all three cFDR-GWASs was obtained by merging any lead SNPs within 250 kb into the same locus represented by the most significant lead SNP.

## Estimation of the number of expected false discoveries

The expected number of false discoveries at a certain cFDR threshold was calculated as the sum of cFDR-values lower than the threshold among the independent lead SNPs of each cFDR-GWAS separately. This is possible since the conditional FDR is defined as a (posterior) probability and can therefore be treated as such. Next, the resulting three numbers were summed to yield a total number of expected false discoveries across all three cFDR-GWASs conducted. This estimate is conservative, given the high degree of dependence between the three cFDR-GWASs, meaning that the true number of false discoveries is lower.

## Gene annotation and overrepresentation analysis

Overrepresentation analysis of GO biological processes and mouse phenotypes among a set of lead SNPs was performed with GREAT[75] v4.0.4 using default settings and a binomial test. Terms were considered significantly overrepresented among the set of lead SNPs at a 5% FDR. For each locus, up to two genes were annotated: the gene with the nearest transcription start site (TSS) centromeric from the lead SNP and the gene with the nearest TSS telomeric from the lead SNP on the condition that their TSSs were within 1 Mb of the lead SNP or within a curated regulatory region. For loci where none of the annotated genes were strongly supported by literature, a gene was manually annotated if it was within 1 Mb, had strong literature support (e.g., involved in craniofacial syndromes or with experimental evidence of involvement in craniofacial development), and was located in the same topologically associated domain based on micro-C in human embryonic stem cells[141].

## REVIGO analysis

The list of GO biological processes used for REVIGO[76] analysis was compiled by taking the union set of terms that were overrepresented (5% FDR) using the binomial test in GREAT among the loci (1% cFDR) from each cFDR-GWAS separately. A semantic space was then generated using REVIGO[76] v1.8.1 with the SimRel[142] similarity measure, which was the default option, and the list size set to "Large (0.9)". Coordinates of terms in the obtained space are available from Supplementary Data 7.

For visualization purposes, terms were broadly categorized in three groups and "other". Any term that contained "regulation" was assigned to the "regulation" category. Other terms that contained either "signal" or "response" were assigned to the "signaling group" and those that contained either "development", "differentiation", "formation", or "-ogenesis" were assigned to the "development / morphogenesis / differentiation" group. All other terms were considered "other". The final plots were made in Matlab 2023a.

## STRING analysis

Gene-interaction networks were constructed using STRING[143] v12.0 using all interaction sources combined (text mining, experiments, databases, co-expression, neighborhood, gene fusion, co-occurrence; default) and a medium confidence cutoff (0.4; default). For added flexibility in visualizing the network, the final plots were made in Matlab 2023a based on the node coordinates and edge weights from the STRING exports.

## Stratified LD-score regression

GWAS summary statistics for the cranial vault shape GWAS in the European subset ($n = 4198$) of the ABCD cohort were obtained from our previous work[6]. LD scores were created for each annotation (corresponding to a set of differential or control distal ATAC-seq peaks) using the 1000 Genomes Phase 3 population reference[138]. Each annotation's heritability enrichment was computed by adding the annotation to the baseline LD model and regressing against trait chi-squared statistics using HapMap3 SNPs with the S-LDSC[135] package v.1.0.1. TWIST1-dependent ATAC-seq peaks, as well as all ATAC-seq peaks in CNCCs, were obtained from Kim et al.[83] We note that the TWIST1-dependent peak sets span 0.67% and 0.73% of SNPs for acute depletion and long-term loss, respectively (based on 1000 Genomes SNP annotation in individuals of European ancestry, which encompass our GWAS populations), above the 0.5% defined as a large annotation.

## LocusZoom plots

Summary statistics of relevant GWASs[6,7,33] were obtained from GWAS Catalog (GCST90270327; GCST90012880; and GCST90007266). LD with the lead SNP was obtained using the NIH LDlink[144] ("LD proxy") rest API, based on the 1000 Genomes Phase 3 GRCh38 High Coverage EUR genomes[138]. Protein coding genes including their exons were annotated using NCBI RefSeq annotations (available at: http://hgdownload.soe.ucsc.edu/goldenPath/hg19/bigZips/genes/hg19.ncbiRefSeq.gtf.gz). Plots were made in Matlab 2023b.

## Reporting summary

Further information on research design is available in the Nature Portfolio Reporting Summary linked to this article.

## Data availability

All the data and detailed information for the ABCD Study, including MRI scans, genetic markers, and covariates are available under restricted access through the ABCD data repository (https://nda.nih.gov/abcd/) upon completion of the relevant data use agreements. The ABCD data repository grows and changes over time and the data used in this work came from data release 3.0 (https://doi.org/10.15154/1519007 and https://doi.org/10.15154/wthp-7h18). The NYGC 30 × 1000 genomes phased dataset and HGDP dataset are freely available online (http://ftp.1000genomes.ebi.ac.uk/vol1/ftp/data_collections/1000G_2504_high_coverage/working/20201028_3202_phased/, and https://ftp.sra.ebi.ac.uk/1000g/ftp/data_collections/HGDP/data/). The LD block coordinates used in this study are available from Berisa et al.[137] at (https://bitbucket.org/nygcresearch/ldetect-data/src/master/). Mesh templates used for surface registration are available from the FigShare repositories of previous works (https://doi.org/10.6084/m9.figshare.c.6858271.v1[145], https://doi.org/10.6084/m9.figshare.7649024.v1[146], and https://doi.org/10.6084/m9.figshare.c.5089841.v1[147]). Summary statistics for the cFDR-GWASs are available from FigShare (https://doi.org/10.6084/m9.figshare.c.7680035.v1[148]). An overview of PubMed IDs and URLs for the GWAS summary statistics used in this work are provided in Supplementary Table 1. Transcription factor binding site coordinates can be obtained from the TF Link database v1.0 (https://tflink.net/download/, and https://cdn.netbiol.org/tflink/download_files/TFLink_Homo_sapiens_bindingSites_All_annotation_v1.0.tsv.gz). A list of genomic regions differentially accessible upon TWIST1 loss or depletion can be obtained from Kim et al.[83] (GEO: GSE230319). RefSeq gene and exon annotations used in LocusZoom plots are freely available from the UCSC golden path (http://hgdownload.soe.ucsc.edu/goldenPath/hg19/bigZips/genes/hg19.ncbiRefSeq.gtf.gz). The Source data behind the graphs in the paper can be found in Supplementary Data 1–8.

## Code availability

KU Leuven provides the MeshMonk v.0.0.6 spatially dense facial-mapping software, free to use for academic purposes available at (https://github.com/TheWebMonks/meshmonk). The latest version is available from the FigShare repository of a previous publication (https://doi.org/10.6084/m9.figshare.c.6858271.v1[145]). Matlab implementations of the hierarchical spectral clustering to obtain facial segmentations are available from a previous publication (https://doi.org/10.6084/m9.figshare.7649024.v1[146]). The conditional FDR software is available from Github (https://github.com/precimed/pleiofdr). The statistical analyses in this work were based on functions in Matlab 2021a–2023b, python v3.7.6, R v4.2.1, PLINK 2.0, bcftools v1.10.2, vcftools v0.1.17, SHAPEIT v4.2.2, IMPUTE5 v1.1.5, imp5Chunker v1.1.5, ADMIXTURE v1.3.0, MeshMonk v0.0.6, GREAT v4.0.4, FUMA v1.6.1, SimpleITK v 2.1.0, REVIGO v1.8.1, FreeSurfer v6.0.0, StringDB v12.0, LDSC v1.0.1

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

## Acknowledgements

Data used in the preparation of this article were obtained from the Adolescent Brain Cognitive Development (ABCD) Study (https://abcdstudy.org), held in the NIMH Data Archive (NDA). This is a multisite, longitudinal study designed to recruit more than 10,000 children age 9–10 and follow them over 10 years into early adulthood. The ABCD Study is supported by the National Institutes of Health and additional federal partners under award numbers U01DA041048, U01DA050989, U01DA051016, U01DA041022, U01DA051018, U01DA051037, U01DA050987, U01DA041174, U01DA041106, U01DA041117, U01DA041028, U01DA041134, U01DA050988, U01DA051039, U01DA041156, U01DA041025, U01DA041120, U01DA051038, U01DA041148, U01DA041093, U01DA041089, U24DA041123, U24DA041147. A full list of supporters is available at https://abcdstudy.org/federal-partners.html. A listing of participating sites and a complete listing of the study investigators can be found at https://abcdstudy.org/consortium_members/. ABCD consortium investigators designed and implemented the study and/or provided data but did not necessarily participate in analysis or writing of this report. This manuscript reflects the views of the authors and may not reflect the opinions or views of the NIH or ABCD consortium investigators. The resources and services used in this work were provided by the VSC (Flemish Supercomputer Center), funded by the Research Foundation—Flanders (FWO) and the Flemish Government. This project was supported in part by the National Institute of Dental and Craniofacial Research: R01DE027023 (S.M.W., J.R.S., P.C., J.W.) and R00DE032729 (S.N.).

## Author contributions

S.G. and P.C. conceptualized the study. S.G., H.H., N.H., M.Y., P.C., S.W., and S.N. carried out the data curation. S.G. and S.N. carried out the formal analysis. S.G., S.N., M.Y., J.W., and P.C. carried out the investigation. S.G. did the visualization. P.C., S.M.W, S.W., J.W., J.R.S., and S.N. were responsible for funding acquisition. P.C., J.W., S.M.W., and S.W. carried out the supervision. S.G. wrote the original draft. S.G., H.H., M.Y., S.N., N.H., J.R.S., M.D.S., J.W., S.W., S.M.W. and P.C. reviewed and edited the final manuscript.

## Competing interests

The authors declare no competing interests.

## Additional information

[1]Department of Human Genetics, KU Leuven, Leuven, Belgium. [2]Medical Imaging Research Center, University Hospitals Leuven, Leuven, Belgium. [3]Department of Chemical and Systems Biology, Stanford University School of Medicine, Stanford, CA, USA. [4]Departments of Genetics and Biology, Stanford University School of Medicine, Stanford, CA, USA. [5]Division of Gastroenterology, Hepatology, and Nutrition, Boston Children's Hospital, Boston, MA, USA. [6]Department of Pediatrics, Harvard Medical School, Boston, MA, USA. [7]Department of Electrical Engineering, ESAT/PSI, KU Leuven, Leuven, Belgium. [8]Department of Cell Biology & Anatomy, Cumming School of Medicine, Alberta Children's Hospital Research, Institute, University of Calgary, Calgary, AB, Canada. [9]Department of Biology, Indiana University Indianapolis, Indianapolis, IN, USA. [10]Center for Craniofacial and Dental Genetics, Department of Oral and Craniofacial Sciences, University of Pittsburgh, Pittsburgh, PA, USA. [11]Department of Anthropology, Pennsylvania State University, State College, PA, USA. [12]Department of Human Genetics, University of Pittsburgh, Pittsburgh, PA, USA. [13]Department of Anthropology, University of Pittsburgh, Pittsburgh, PA, USA. [14]Department of Developmental Biology, Stanford University School of Medicine, Stanford, CA, USA. [15]Howard Hughes Medical Institute, Stanford University School of Medicine, Stanford, CA, USA. [16]Murdoch Children's Research Institute, Melbourne, VIC, Australia. ✉e-mail: seppe.goovaerts@kuleuven.be; peter.claes@kuleuven.be

