## [Transparent Peer Review file · Communications Biology]

Enhanced insights into the genetic architecture of 3D cranial vault shape using pleiotropy-informed GWAS

Corresponding Author: Mr Seppe Goovaerts

This manuscript has been previously reviewed at another journal. This document only contains information relating to versions considered at Communications Biology.

Version 0:

Reviewer comments:

Reviewer #1

(Remarks to the Author)

In this paper, the authors investigate the genetic underpinnings of cranial vault shape by leveraging data from related traits using a pleiotropy-informed conditional GWAS approach. Using summary statistics from ABCD and UKBiobank, they examine the additional yield of the significant loci when conditioning the cranial vault results on other related traits, including brain, face, and BMD. The authors found 3 fold of increase of the number of significant loci based on the conditional FDR at 1% threshold and some evidence indicating the differential enrichments. In general, this paper is well written by a group focused on the cranial facial development. Although the findings are relatively incremental, considering their previous findings on face GWAS and cranial vault GWAS, the attempts to further elucidate their inter-dependent relationships are worth to be noted. Here are some suggestions and comments hopefully can enhance this manuscript further:

1. It is nice to see the authors put the increased yield in context, discussing the additional loci found across different method, i.e. 1% FDR, 5%FDR, and unconditioned GWAS with more lenient threshold. However, since most of the loci are overlapped, it is unclear if the conditional GWAS has bring in the trait specificity beyond the boost on the discovery power. It would be great if the authors can demonstrate the conditional analyses can provide specific information about the newly discovered loci.
2. The variance explained analyses are performed using several different sample compositions. The variance explained by cranial vault and brain are based on a sample much larger than the variance explained by cranial vault and face. It renders the comparisons difficult, as the difference in the variance explained might also driven by the uncertainty in the model fit due to limited sample sizes. Similarly, the characteristics of the samples size and the sample compositions for each summary statistic are not well described in the beginning, making it hard to interpret the results.
3. The argument based on the increased variance explained when more loci are discovered is on a shaky ground because the variance explained is for sure to increase when you include more loci into the calculation. The variance is strictly positive, so the addition would monotonically increase the variance explained.
4. The cranial vault GWAS seems to be based on the diverse ABCD samples but many of the LD based analyses and discussions are based on the European reference. This mismatch is weakening the arguement the authors put forward.
5. The ordering of the subfigures are not following specific logic but seem to base on the size of the subfigures. For example, in the figure 1, the subfigures start with vertical arrangement, but then shift to horizontal arrangement after f. It makes the figures hard to follow. Similarly, Figure 5 combined two different analyses into single one figure yet the subfigure flows interspersed. It took me awhile to realize we are looking at two different loci.

Reviewer #2

(Remarks to the Author)

In this manuscript the authors use conventional methods to re-analyse recently published imaging and GWAS data. By doing so, the authors present interesting findings in an understudied area. Overall, the manuscript is very well written, with the premise of the study outlined clearly and the findings well described. The figures and data presented are clear and easy to read and support the authors claims. They have cited relevant studies accordingly, describing how this study complements previously published literature. The authors also address the limitations of the study. The methods are documented clearly so that replication can be performed.

Reviewer #3

(Remarks to the Author)

This manuscript by Goovaerts et al. explores the ABCD cohort to perform a conditional GWAS of cranial vault shape, conditioning on known genetic associations with other craniofacial features such as face shape and brain shape. As is expected from their shared biological development, the study argues that genetic effects are shared between cranial vault shape and face and brain shape. They therefore perform a conditional GWAS to present additional associations that were not observed in their previous unconditional GWAS in the same cohort [ref 6].

1. As a peer who works in this area but has not used exactly the same tools such as conditional FDR, I found the manuscript's methods really difficult to follow through if I were to replicate the same analysis in a different dataset. From what I understand, the conditional GWAS is the core message of the paper. But there is not a lot of methodological detail on this core aspect, as the reader is mostly redirected to ref 44. But there are a lot of computational steps (Estimation of the number of expected false discoveries, FUMA, GREAT, REVIGO etc.) that were used, and a lot of way-arounds (such as genomic Spearman correlations) which obscure the core aspect to a certain extent.

2. As the manuscript explains, the assertion that forms the basis of the conditional GWAS is expected from their shared biological development. All parts of the human face form during the embryonic stage from a set of pharyngeal arches, and quite a lot is known about their genetic basis, not only from humans but from various animals; e.g. that different genes are involved in different sets of pharyngeal arches, and consequently, different blocks of the human face are influenced by different sets of genes. The manuscript doesn't discuss about the broad state of current knowledge in the genetic basis of the development of cranial vault, face, and brain shape in the introductions; the manuscript seems to be too rooted in computation but too little in fundamental biology.

3. It is generally well known in GWAS literature that a conditional approach, either using selected SNPs informed from an association study, or using a polygenic score constructed from the results of an association study, can be used to improve power and aid discovery of additional SNPs [e.g. Campos et al. Nat Genet 2023].

4. It is also relevant to note that the authors have decided here to use an FDR-based method to account for multiple comparisons and control type I error. Their previous unconditional GWAS in the same cohort [ref 6] used the traditional GWAS convention of a significance threshold ($5e-8$) that controls for the FWER, which is a much more stringent procedure than FDR. There is nothing inherently wrong with using the FDR, which is a legitimate statistical procedure, but is a much more lenient one – if they had used FDR in the original unconditional GWAS they would've identified many more associations as significant anyway.

Considering these two points, I am finding it difficult to see a substantial methodological novel contribution in this manuscript.

5. In parallel, I am confused by some of the methodological decisions made, which probably adds to the difficulty of following the manuscript. For example, both individual-level genotype data and individual-level phenotype data, of cranial vault, face, and brain shape (or the PCs derived from them), are available in the ABCD cohort. Thus, they could've directly used the individual-level data to perform the conditional GWASes, which would've simplified some of the computational way-arounds that they had to attempt. But I am not sure why they don't do so. E.g. in the methods they explain that despite having access to imputed genotype data, "the conditional GWASs were run using summary statistics from prior works".

Sometimes summary statistics from other cohorts are preferred if there are several orders of magnitude difference in sample sizes, since the available summary statistics are more precisely estimated, rather than conducting a small GWAS in the available cohort. But that doesn't seem to be the case here.

6. Other methodological details are also difficult to follow. As explained in the methods and the previous study [ref 6], the ABCD cohort is a mixed- or multi-ethnic cohort. However, the study attempts to extract a subset of European individuals [line

549], uses LD blocks estimated only in European populations [line 657], or use annotations based on European ancestry [line 671].

7. In results, a section presents that some of the fold enrichment of statistical association can be explained by genomic Spearman correlations. But it is not clear from the methods how or to what extent this can be explained.

Overall, I do not intend to suggest that the methodological decisions made in the manuscript are necessarily incorrect. But they are sometimes difficult to follow and sometimes difficult to perceive as appropriate.

Reviewer #4

(Remarks to the Author)

Version 1:

Reviewer comments:

Reviewer #1

(Remarks to the Author)

The authors have nicely addressed all my questions. I have no more comments on this manuscript.

Reviewer #3

(Remarks to the Author)

The authors have provided detailed clarifications on several comments through their responses and changes in the manuscript; their efforts are much appreciated. There were multiple aspects of their methodological approach which were not clear earlier but are now easier to understand.

This improved understanding however allows me to ask some further questions about their methodology, and suggest some further changes.

Major comment:

In response to reviewer 1 point 2 about the variance explained analysis, the authors mention "... have now rerun the former analysis on a single sample for which we had high quality brain, face, and vault data available." This is very relevant for my major point 5 where I was requesting the authors to run a direct conditional analysis using the raw data in the ABCD cohort. I don't find the authors' arguments against doing this particularly strong.

As far as I still understand, the core message of the paper is the methodology that conditioning on those additional craniofacial traits improves gene discovery for vault shape. Other methodology, such as conditional FDR, are used to assist this aim, but is not the core message itself; therefore the study does not need to be wedded to cFDR as the only conditioning method, especially since it is a rather complex and indirect method, with various approximations, and therefore not widely used either.

I understand that BMD data will not be available in the ABCD cohort, but conditioning on the brain and face can still be done to directly demonstrate the improvement brought about by conditioning.

I also understand that the sub-sample with all three kinds of data is somewhat smaller (2500 compared to 4200). But when comparing the unconditional vs. conditional results within the same 2500, this isn't necessarily a problem. They can consistently use the FDR or the 5E-8 threshold for both.

And since the main cohort's sample size is fixed, while I understand that the summary statistics for the conditioning cohorts are from larger sample sizes, I don't think it's guaranteed that a moderate increase in their sample size necessarily beats the advantage of working with raw data.

So, as they are able to use the subset of the ABCD cohort for which they have high quality brain, face, and vault data for the variance analysis, I think it is really important to do the direct conditional analysis on this subset too, and make a more direct comparison.

Minor comments:

1. The authors have helpfully pointed out the comparisons where a 1% FDR version of the original GWAS is used. I believe this should be the main basis for comparison, not the 21 loci initially reported using the 5E-8 threshold. For simplicity and ease of understanding, I suggest removing comparisons to the original list of loci throughout.

2. For the enrichment analysis in Figure 4, a 5% FDR version seem to have been used instead of 1% FDR. While the

authors mention 'additional loci', I am not sure whether the enrichment analysis is performed on the loci which are present in the 5% CFDR analysis but not in the 5% FDR version of the unconditional GWAS? I think that'll be the most appropriate set of loci to check enrichment on.

3. Among the new additions to the manuscript, I find this paragraph difficult to mechanistically understand.

"Still, we noted a few examples where the choice of conditional trait was directly related to the findings. Only when conditioning on BMD, we identified a locus near LRP5 at 1% FDR, for which loss-of-function mutations are associated with osteoporosis and gain-of-function mutations with higher bone mass and craniosynostosis. Additionally, a locus near MSX1 was only identified at 1% FDR upon conditioning on facial shape. This gene plays a critical role in facial development, including the frontal bone, with mutations linked to multiple craniofacial anomalies such as cleft lip and palate."

If there was a gene associated with bone mass in the unconditioned analysis, which goes away when conditioning on BMD, I'd understand that the initial association was demonstrating an indirect effect of the gene on bone mass which goes away when conditioning on a trait capturing bone mass. But here the opposite seems to happen. What might be the reason for this?

4. The abstract and introduction should directly mention which traits have been conditioned on. Currently indirect phrasing is used, e.g. "these auxiliary traits were then leveraged".

5. The authors now clarify that only the European sub-sample of the ABCD data was used for analysis. This needs to be clearly stated in the Methods under 'GWAS summary data'.

6. While the authors have provided some further details on the Conditional FDR method, some further details on the genomic Spearman correlations method is needed as well.

7. "Unsurprisingly, genomic Spearman correlations were good at predicting cross-trait enrichments of association ($R^2 = 0.685$; $P = 8.91e-4$)." The methodology is still not clear – how does one predict cross-trait enrichments of association from a genomic Spearman correlation?

8. The authors have now added a summary of the biological processes behind craniofacial development. This is helpful but doesn't mention their known genetic drivers at all. E.g. do some of the major known genes affect more than one of cranial vault, face, and brain shape?

Version 2:

Reviewer comments:

Reviewer #3

(Remarks to the Author)

In replying to my major point, I'm sure it became abundantly clear to the authors that this reviewer had completely misunderstood what 'conditional GWAS' meant in their context! I presume they've been rather exasperated, although they've been extremely polite in their replies, and their current response clarifies this confusion substantially.

It might be useful to acknowledge though, that many other readers would come from a background like mine where 'conditional GWAS' is known to refer to the kind of adjustment-based regression approach used in GCTA-COJO (Yang et al. Nature Genetics 2012) and also in other quantitative genetics approaches such as Conditional eQTL (Dobyn et al. AJHG 2018). While one can see that the word 'conditional' in the context of FDR arises from conditional probabilities, converting it to 'conditional GWAS' in this manuscript seems to have started the confusion. I note that this phrase was not used in the original paper [ref 54] which proposed the cFDR procedure.

Perhaps the authors can consider changing the phrase. The word 'conditional' in front of GWAS in the title is perhaps unnecessary as 'pleiotropy-informed GWAS' is clear enough. But generally they might want to find a better name for this GWAS procedure, something like 'conditional FDR-based GWAS' or 'cFDR-GWAS' throughout the manuscript should help to clear up the confusion.

The authors have provided additional clarifications on Minor point 2. I have one further question: the enrichment analysis seems to have been done on all loci that were significant (at 5%) in the cFDR-based GWAS. As they are talking about 'additional loci', I would expect that this enrichment analysis would be done on loci excluding those found at 5% FDR in the original GWAS? Can this be clarified please.

Reviewer #1 (Remarks to the Author):

In this paper, the authors investigate the genetic underpinnings of cranial vault shape by leveraging data from related traits using a pleiotropy-informed conditional GWAS approach. Using summary statistics from ABCD and UKBiobank, they examine the additional yield of the significant loci when conditioning the cranial vault results on other related traits, including brain, face, and BMD. The authors found 3 fold of increase of the number of significant loci based on the conditional FDR at 1% threshold and some evidence indicating the differential enrichments. In general, this paper is well written by a group focused on the cranial facial development. Although the findings are relatively incremental, considering their previous findings on face GWAS and cranial vault GWAS, the attempts to further elucidate their inter-dependent relationships are worth to be noted. Here are some suggestions and comments hopefully can enhance this manuscript further:

We thank the reviewer for their helpful comments and suggestions.

1. It is nice to see the authors put the increased yield in context, discussing the additional loci found across different method, i.e. 1% FDR, 5%FDR, and unconditioned GWAS with more lenient threshold. However, since most of the loci are overlapped, it is unclear if the conditional GWAS has bring in the trait specificity beyond the boost on the discovery power. It would be great if the authors can demonstrate the conditional analyses can provide specific information about the newly discovered loci.

This is an interesting question. On a broader level, we mostly observed results that were highly similar across all three conditional GWASs and the unconditioned GWAS. For example, the lead SNPs from each respective GWAS explained very similar patterns of shape variance and yielded very similar GO processes as illustrated in Fig3. We noted only subtle differences, e.g., that the facial conditional analysis yielded lead SNPs that were slightly biases towards explaining variance around the forehead.

We took this opportunity to look further into individual loci that were uniquely identified in only one of the conditional analyses and added two specific examples to the discussion for which we believe the conditioning trait was a determining factor in the discovery:

“Still, we noted a few examples where the choice of conditional trait was directly related to the findings. Only when conditioning on BMD, we identified a locus near LRP5 at 1% FDR, for which loss-of-function mutations are associated with osteoporosis and gain-of-function mutations with higher bone mass and craniosynostosis. Additionally, a locus near MSX1 was only identified at 1% FDR upon conditioning on facial shape. This gene plays a critical role in facial development, including the frontal bone, with mutations linked to multiple craniofacial anomalies such as cleft lip and palate.”

While doing this, we noticed that most of these unique loci reside closely near the 1% FDR threshold and found that 28/51 (54.9%) of them were also identified by at least one other conditional GWAS at a 5% FDR cutoff, complicating the interpretation trait-specific loci. This has now been added to the Results:

“These loci strongly overlapped across all three conditional GWAS analyses (Fig 3a). Furthermore, among the 51 loci identified exclusively in a single conditional GWAS at a 1% cFDR threshold, 28 (54.9%) also achieved a cFDR < 5% in at least one of the other analyses (Supplementary Data 1).”

2. The variance explained analyses are performed using several different sample compositions. The variance explained by cranial vault and brain are based on a sample much larger than the variance explained by cranial vault and face. It renders the comparisons difficult, as the difference in the variance explained might also be driven by the uncertainty in the model fit due to limited sample sizes. Similarly, the characteristics of the sample size and the sample compositions for each summary statistic are not well described in the beginning, making it hard to interpret the results.

We acknowledge this concern and have now rerun the former analysis on a single sample for which we had high quality brain, face, and vault data available. As such, the analyses use the same sample sizes thereby making the results easier to interpret.

To increase transparency for the analysis shown in Fig1c, we added the sample sizes for each GWAS study to Supplementary Table 1, as well as to the caption of Fig 1. We also acknowledge that the results shown in Fig1c may be difficult to interpret because of difference in sample size. However, despite this, we show in Fig1d that consistent estimates of the slope are obtained when using GWAS data with constant sample size, demonstrating robustness of the initial estimate. This is now more clearly mentioned in the text:

“We also note that despite the variability in sample size in Fig 1c, we obtained consistent estimates of the slope when using GWAS data with constant sample sizes (Fig 1d), demonstrating robustness of the initial estimate.”

3. The argument based on the increased variance explained when more loci are discovered is on a shaky ground because the variance explained is for sure to increase when you include more loci into the calculation. The variance is strictly positive, so the addition would monotonically increase the variance explained.

To address this, we have added a panel to Fig3 to show the distribution of the phenotypic variance explained by individual lead SNPs in relation to a simulated null distribution based on 1000 permutations.

Additionally, when talking about the loci identified at 5% FDR, we no longer use increased phenotypic variance explained as an argument, but just note it. The following wording was replaced:

“we lowered the significance threshold to 5% cFDR in each conditional GWAS and subsequently merged their genomic loci into 328 independent loci. ... we reasoned that the additional loci identified would be expected to be enriched for biologically meaningful information. This is evidenced by the 8.53% of overall cranial vault shape explained by the 328 lead SNPs, more than doubling what the initial set of lead SNPs explained. Furthermore, most of the top 20 GO biological processes identified for the initial 120 loci were more strongly represented among the broader set of 328 loci”

by:

“we lowered the significance threshold to 5% cFDR in each conditional GWAS and subsequently merged their genomic loci into 328 independent loci, explaining 8.53% of overall cranial vault shape. ..., we reasoned that the additional loci identified would be expected to be enriched for biologically meaningful information. For example, most of the top 20 GO biological processes identified for the initial 120 loci were more strongly represented among the broader set of 328 loci”

4. The cranial vault GWAS seems to be based on the diverse ABCD samples but many of the LD based analyses and discussions are based on the European reference. This mismatch is weakening the argument the authors put forward.

We acknowledge this point by the reviewer and added several clarifications to the text.

Most of the GWAS summary statistics were obtained from a predominantly European ancestry sample. This includes the ABCD cranial vault shape GWAS, for which we have previously inferred that ~80% of the alleles in the sample were derived from recent European ancestors. This was added to the methods:

“GWAS summary statistics were obtained from our recent cranial vault shape GWAS6 and from recent, large-scale GWAS studies7,28,60–68 on other complex phenotypes conducted in independent cohorts of predominantly recent European ancestry. This includes the cranial vault GWAS in the ABCD cohort, for which ~80% of the alleles were derived from recent European ancestors.”

In our work, we mostly used LD to establish a set of independent SNPs (condFDR analysis, peak detection, genomic Spearman correlations). Because the LD structure in European populations is less fine-grained than in African populations, the assumptions of independence should still hold, in contrast to the inverse scenario. Since the condFDR analysis and genomic Spearman correlations also use a secondary dataset, derived from a European-ancestry sample, we believe that using European LD for these analyses is appropriate to minimize the risk of false positives.

This was briefly added to the methods:

“We used European-derived LD blocks since all GWAS data were generated in samples of predominant recent European ancestry.”

Furthermore, S-LDSC was performed on the GWAS summary statistics from only the European subset of the ABCD, together with European annotations. This version of the GWAS is freely available from previous publication. This decision was motivated by the issues we encountered when running LDSC on the multi-ancestry GWAS statistics, even when using ancestry-adjusted LD-scores. This is now clearly indicated in the text:

“GWAS summary statistics for the cranial vault shape GWAS in the European subset (n = 4198) of the ABCD cohort were obtained from our previous work.”

5. The ordering of the subfigures are not following specific logic but seem to base on the size of the subfigures. For example, in the figure 1, the subfigures start with vertical arrangement, but then shift to horizontal arrangement after f. It makes the figures hard to follow. Similarly, Figure 5 combined two different analyses into single

one figure yet the subfigure flows interspersed. It took me awhile to realize we are looking at two different loci.

We thank the reviewer for this point and have re-ordered the panels, wherever possible, to make the figures easier to follow. For example, the panels in Fig1 are now organized in two rows, organized as a -> b -> c, then d -> e -> f -> g. Similarly, the panels in Fig5 are organized in two columns as a, then b -> c -> d.

Reviewer #2 (Remarks to the Author):

In this manuscript the authors use conventional methods to re-analyse recently published imaging and GWAS data. By doing so, the authors present interesting findings in an understudied area. Overall, the manuscript is very well written, with the premise of the study outlined clearly and the findings well described. The figures and data presented are clear and easy to read and support the authors claims. They have cited relevant studies accordingly, describing how this study complements previously published literature. The authors also address the limitations of the study. The methods are documented clearly so that replication can be performed.

We thank the reviewer for their review and their positive feedback on the manuscript.

Reviewer #3 (Remarks to the Author):

This manuscript by Goovaerts et al. explores the ABCD cohort to perform a conditional GWAS of cranial vault shape, conditioning on known genetic associations with other craniofacial features such as face shape and brain shape. As is expected from their shared biological development, the study argues that genetic effects are shared between cranial vault shape and face and brain shape. They therefore perform a conditional GWAS to present additional associations that were not observed in their previous unconditional GWAS in the same cohort [ref 6].

We thank the reviewer for their helpful comments and suggestions.

1. As a peer who works in this area but has not used exactly the same tools such as conditional FDR, I found the manuscript's methods really difficult to follow through if I were to replicate the same analysis in a different dataset. From what I understand, the conditional GWAS is the core message of the paper. But there is not a lot of methodological detail on this core aspect, as the reader is mostly redirected to ref 44. But there are a lot of computational steps (Estimation of the number of expected false discoveries, FUMA, GREAT, REVIGO etc.) that were used, and a lot of way-arounds (such as genomic Spearman correlations) which obscure the core aspect to a certain extent.

We thank the reviewer for bringing this to our attention. To make the Methods more focused on the core aspect of the paper, we 1) added more details on the conditional FDR method and 2) moved some of the elaborate imaging and genotype processing steps to the Supplementary Note.

2. As the manuscript explains, the assertion that forms the basis of the conditional

GWAS is expected from their shared biological development. All parts of the human face form during the embryonic stage from a set of pharyngeal arches, and quite a lot is known about their genetic basis, not only from humans but from various animals; e.g. that different genes are involved in different sets of pharyngeal arches, and consequently, different blocks of the human face are influenced by different sets of genes. The manuscript doesn't discuss about the broad state of current knowledge in the genetic basis of the development of cranial vault, face, and brain shape in the introductions; the manuscript seems to be too rooted in computation but too little in fundamental biology.

We acknowledge this point and added a paragraph to introduce the shared developmental basis of the brain and cranium:

“Formation of the human head starts early in development when the rostral end of the neural tube forms the hindbrain, midbrain, and forebrain, the latter later developing into the cerebral hemispheres⁸. From the same neural tube region, cranial neural crest cells (CNCCs) delaminate and migrate ventrally^{9,10}. Guided by positional cues, anterior-most CNCCs form the frontonasal skeleton, while posterior CNCCs populate the pharyngeal arches to form the bone and cartilage of the jaws¹⁰. The rate of growth of the early brain influences the positioning of facial structures¹¹, while the flat bones of the neurocranium, derived from the paraxial mesoderm, are joined by flexible sutures that accommodate brain expansion¹². Dysregulated coordination between the brain and craniofacial mesenchyme results in congenital malformations such as cleft lip and palate or craniosynostosis^{11–14}. This underscores the importance of considering their shared development when studying craniofacial variation.”

3. It is generally well known in GWAS literature that a conditional approach, either using selected SNPs informed from an association study, or using a polygenic score constructed from the results of an association study, can be used to improve power and aid discovery of additional SNPs [e.g. Campos et al. Nat Genet 2023].

We thank the reviewer for this suggestion. The updated manuscript now mentions this briefly in the Discussion and refers to the suggested reference and to other alternative strategies:

“While additional data sources and larger sample sizes are necessary to further elucidate the genetic architecture of complex traits in general, conditional GWAS strategies already enhance genetic insights in the meantime^{49,123–125}. In this work, we specifically opted for the conditional FDR method as it could seamlessly deal with the multivariate GWAS statistics of cranial vault shape and the other morphological phenotypes.”

4. It is also relevant to note that the authors have decided here to use an FDR-based method to account for multiple comparisons and control type I error. Their previous unconditional GWAS in the same cohort [ref 6] used the traditional GWAS convention of a significance threshold ($5e-8$) that controls for the FWER, which is a much more stringent procedure than FDR. There is nothing inherently wrong with using the FDR, which is a legitimate statistical procedure, but is a much more lenient one – if they had used FDR in the original unconditional GWAS they would've identified many more associations as significant anyway.

In the original version of the manuscript, we already addressed this valid point by applying an FDR correction to the unconditional GWAS. At the same 1% FDR threshold as the conditional GWAS, we identified 46 independent loci. This is more than the 21 loci at $P < 5e-8$, but less than the 57, 83, and 92 loci identified in each of the conditional GWASs (120 after merging).

We believe this is clearly indicated in the results: *“In total, we identified 120 genomic loci associated with cranial vault shape at a 1% cFDR in at least one of the conditional GWAS ... a marked increase compared to the 21 loci initially reported in the unconditioned GWAS. Even when re-evaluating the number of loci identified at a 1% FDR in the original unconditioned GWAS (instead of at a P-value threshold of $< 5e-8$), only 46 loci were significant... Notably, the GWAS conditioned on brain and facial shape yielded a higher number of genetic loci ($n = 92$ and $n = 83$ respectively) compared to the one conditioned on BMD ($n = 57$)”*

Considering these two points, I am finding it difficult to see a substantial methodological novel contribution in this manuscript.

We would like to re-emphasize that the manuscript is not intended to contribute methodology, but instead aims to address the data scarcity that exists within the field of craniofacial genetics. While this scarcity has always existed, conditional analyses are seldomly applied in this field despite being promising.

5. In parallel, I am confused by some of the methodological decisions made, which probably adds to the difficulty of following the manuscript. For example, both individual-level genotype data and individual-level phenotype data, of cranial vault, face, and brain shape (or the PCs derived from them), are available in the ABCD cohort. Thus, they could've directly used the individual-level data to perform the conditional GWASes, which would've simplified some of the computational work-arounds that they had to attempt. But I am not sure why they don't do so. E.g. in the methods they explain that despite having access to imputed genotype data, “the conditional GWASs were run using summary statistics from prior works”.

Sometimes summary statistics from other cohorts are preferred if there are several orders of magnitude difference in sample sizes, since the available summary statistics are more precisely estimated, rather than conducting a small GWAS in the available cohort. But that doesn't seem to be the case here.

We acknowledge that cross-trait analyses can also be conducted using data from the same set of individuals and agree that this could be an interesting avenue for future work, for example, by segmenting multiple structures from the same MRI. Currently, however, the abundance of artefacts and soft tissue compression in the ABCD MRIs results in low sample sizes for genetic investigations and thus limits this idea, especially when requiring that both the facial and cranial vault surface are of high quality ($n \sim 2500$). In the meantime, published GWAS analyses on brain and facial scans have used much larger sample sizes ($n = 19,644$ and $n = 8246$ respectively), higher quality data in the case of the face, have yielded excellent power, and are freely available.

Additionally, the use of published GWAS allows us to explore traits that were not recorded in the ABCD study, as their efforts have been continually focused on brain development. Notably, bone mineral density (BMD) measurements were not available for this cohort, which we deem key to our investigations, e.g., based on the prior overlap between BMD and craniosynostosis. Only when leveraging BMD we identified *LRP5* and *JAG1*, two genes associated with craniosynostosis. Moreover, since BMD is not a morphological phenotype, it offers a different perspective on our analyses yet converges onto similar results.

From a methodological perspective, we find the conditional FDR to be a good fit for this study as it can deal with multivariate GWAS summary data seamlessly and is an established method. The former is in contrast with most other methods and software packages which are typically not directly applicable to our multivariate GWAS statistics due to their reliance on signed effect size estimates, hence requiring workarounds.

Still, we thank the reviewer for this suggestion and will consider it in future work.

6. Other methodological details are also difficult to follow. As explained in the methods and the previous study [ref 6], the ABCD cohort is a mixed- or multi-ethnic cohort. However, the study attempts to extract a subset of European individuals [line 549], uses LD blocks estimated only in European populations [line 657], or use annotations based on European ancestry [line 671].

We acknowledge this point by the reviewer and added several clarifications to the text.

On the selection of an ancestrally homogeneous sample, we added the following to the Methods:

“Only samples with inferred recent European ancestry were retained as abundant imaging artefacts can disproportionately affect individuals of different ancestries due to differences in hair types and facial structures, increasing the risk of spurious results.”

S-LDSC was performed on the GWAS summary statistics from only the European subset of the ABCD, together with European annotations. This version of the GWAS is freely available from previous publication. This decision was motivated by the issues we encountered when running LDSC on the multi-ancestry GWAS statistics, even when using ancestry-adjusted LD-scores. This is now clearly indicated in the text:

“GWAS summary statistics for the cranial vault shape GWAS in the European subset (n = 4198) of the ABCD cohort were obtained from our previous work.”

We used LD blocks inferred in a European sample to estimate the genomic Spearman correlations because most of the GWAS summary statistic available used a predominantly European ancestry sample. This includes the ABCD cranial vault shape GWAS, for which we have previously inferred that ~80% of the alleles in the sample were derived from recent European ancestors. This is now indicated in the methods:

“GWAS summary statistics were obtained from our recent cranial vault shape GWAS6 and from recent, large-scale GWAS studies7,28,60–68 on other complex phenotypes conducted in independent cohorts of predominantly recent European ancestry. This includes the cranial vault GWAS in the ABCD cohort, for which ~80% of the alleles were derived from recent European ancestors.”

Additionally, because the LD structure in European populations tends to be less fine-grained than in African populations, the genomic correlation estimate would be biased towards zero. This was briefly added to the methods:

“We used European-derived LD blocks since all GWAS data was generated in samples of predominant recent European ancestry.”

7. In results, a section presents that some of the fold enrichment of statistical association can be explained by genomic Spearman correlations. But it is not clear from the methods how or to what extent this can be explained.

Thank you for pointing this out. We rephrased this section to better reflect our rationale and to improve clarity:

“As these enrichments are directly related to the Bayesian principles of the conditional FDR method used later in this work, we aimed to investigate how they corresponded to genomic Spearman correlations, another method for assessing genetic overlap that is applicable to multivariate GWAS. Unsurprisingly, genomic Spearman correlations were good at predicting cross-trait enrichments of association ($R^2 = 0.685$; $P = 8.91e-4$). However, for brain and facial shape, the enrichments were substantially higher relative to their genomic correlations with cranial vault shape.”

Overall, I do not intend to suggest that the methodological decisions made in the manuscript are necessarily incorrect. But they are sometimes difficult to follow and sometimes difficult to perceive as appropriate.

We again thank the reviewer for pointing out where our explanations could be improved, and hope that the additions to the text improve clarity and make the manuscript easier to follow.

Reviewer #1 (Remarks to the Author):

The authors have nicely addressed all my questions. I have no more comments on this manuscript.

We again thank the reviewer for their valuable feedback.

Reviewer #3 (Remarks to the Author):

The authors have provided detailed clarifications on several comments through their responses and changes in the manuscript; their efforts are much appreciated. There were multiple aspects of their methodological approach which were not clear earlier but are now easier to understand. This improved understanding however allows me to ask some further questions about their methodology, and suggest some further changes.

We thank the reviewer for their interest and appreciate their engagement in improving the quality of the manuscript. We are also happy that our previous efforts improved the clarity of the work.

Major comment:

In response to reviewer 1 point 2 about the variance explained analysis, the authors mention "... have now rerun the former analysis on a single sample for which we had high quality brain, face, and vault data available." This is very relevant for my major point 5 where I was requesting the authors to run a direct conditional analysis using the raw data in the ABCD cohort. I don't find the authors' arguments against doing this particularly strong.

As far as I still understand, the core message of the paper is the methodology that conditioning on those additional craniofacial traits improves gene discovery for vault shape. Other methodology, such as conditional FDR, are used to assist this aim, but is not the core message itself; therefore the study does not need to be wedded to cFDR as the only conditioning method, especially since it is a rather complex and indirect method, with various approximations, and therefore not widely used either.

I understand that BMD data will not be available in the ABCD cohort, but conditioning on the brain and face can still be done to directly demonstrate the improvement brought about by conditioning.

I also understand that the sub-sample with all three kinds of data is somewhat smaller (2500 compared to 4200). But when comparing the unconditional vs. conditional results within the same 2500, this isn't necessarily a problem. They can consistently use the FDR or the $5E-8$ threshold for both.

And since the main cohort's sample size is fixed, while I understand that the summary

statistics for the conditioning cohorts are from larger sample sizes, I don't think it's guaranteed that a moderate increase in their sample size necessarily beats the advantage of working with raw data.

So, as they are able to use the subset of the ABCD cohort for which they have high quality brain, face, and vault data for the variance analysis, I think it is really important to do the direct conditional analysis on this subset too, and make a more direct comparison.

We thank the reviewer for sharing their suggestions and concerns.

If we understand correctly, the reviewer's mention of using individual-level data directly refers to conditioning in the context of regression, which would be synonymous with "adjusting for". Such an approach would focus on isolating genetic effects on the cranial vault from those on other structures (or bone density) through statistical adjustment of vault shape for the other traits. Based also on the reviewer's concerns expressed in their third minor comment, we believe that such an approach aligns with what they so kindly suggest. However, we may be wrong, in which case, please correct us and share the method you have in mind, along with some references. We would sincerely appreciate it.

We acknowledge the need for further clarifications and thank the reviewer for guiding us. An overview of the clarifications added to the manuscript can be found at the bottom of our response. Please let us start by clearly stating the main question we aim to answer with the current work. That is "*Can we boost the discovery of SNPs associated with cranial vault shape by leveraging prior biological or genetic information?*". The goal is not *just* to boost power, but to directly use prior information to add additional support, i.e., "evidence" for SNPs that didn't reach genome-wide significance previously. We agree that this question can be approached from different angles and through different methods, however, we feel that our use case leans itself quite naturally to a Bayesian framework. In this case, an FDR can be defined as the probability that a SNP is null given that its P-value is equal to or lower than the observed P-value. Following this definition, we can obtain a "prior" (unconditioned) FDR from an existing GWAS and use auxiliary GWAS data as "evidence" to estimate a "posterior" (conditional) FDR. This posterior FDR is referred to as the conditional FDR by the authors of the original paper as it represents a conditional probability.

We acknowledge that the conditional FDR method can be perceived as complex. For example, we anticipated that readers may wonder how and why auxiliary GWAS data can be used as supporting evidence that a SNP is associated with cranial vault shape. This is something we aimed to address in Fig1 of the manuscript, particularly in Fig1c. To help build this intuition, let's consider two traits that have a very strong genetic overlap — for simplicity, let's say the overlap is close 100%. In this case, any SNP that is associated with trait2 is almost certainly also associated with trait1. Consequently, the association with trait2 provides strong evidence for a positive association with trait1 and the "posterior FDR" decreases w.r.t. the "prior FDR". Similarly, consider a scenario where the two traits have no genetic overlap whatsoever, then the information that a

SNP is associated with trait2 provides zero evidence for (nor against) a positive association with trait1. Evidently, real-world scenarios lie somewhere in the middle and in Fig1c, we demonstrate that this assertion is indeed empirically sustained in a real scenario. We do this by showing that SNPs are more likely to be associated with cranial vault shape when they are also associated with genetically correlated traits.

Moreover, by using the conditional FDR method, it is precisely the genetic overlap between two traits that is directly being exploited, which is a key difference with several alternative approaches (e.g., those that rely on statistical adjustment). In that regard, it is similar to other methods such as MTAG (PMID: 29292387), which also exploits genetic overlap to reveal novel associations using GWAS summary statistics and is widely used (e.g., PMID: 36525587). Interestingly, it is not only the shared biological processes that contribute to this genetic overlap, but also morphological correlations, both of which are extensive in our case (see also Introduction). This makes this class of methods successful in cases like ours, as we elaborate on in the Discussion section. Here, we opted for the conditional FDR method, because it can seamlessly deal with multivariate GWAS statistics (i.e., the lack of signed betas or Z-statistics), unlike most other methods (including MTAG).

We hope that these explanations clarify why our research question leans itself, as we believe, so naturally to a Bayesian framework and why the use of GWAS summary statistics is appropriate and inherent to this class of methods. As we see it, this makes the conditional FDR method not only an adequate method to use, but also an attractive one for our use case. While certainly interesting, a comparison of methods would require a substantial investment of time and changes to the manuscript and is beyond the scope of this work. Still, we do appreciate the reviewer's suggestion of working with the raw data directly and will consider it in future work, as already communicated previously.

To address the concerns of the reviewer, we made several changes to the text with the aim to clarify the terminology, methodology, and rationale.

First, we clearly state our aim in the last paragraph of the Introduction:

“In this study, we aim to leverage prior biological and genetic information to enhance the discovery of genomic loci underlying cranial vault morphology.”

To make the methodology easier to understand and the manuscript easier to follow, we added the following information to the last paragraph of the Introduction:

“Given the extensive genomic overlap between brain, facial, and cranial vault morphology⁶, as well as evidence that genes associated with bone mineral density (BMD) control cranial suture ossification^{52,53}, we demonstrate that single nucleotide polymorphisms (SNPs) associated with those traits have an increased likelihood of association with cranial vault shape. In an empirical Bayesian framework, this can be interpreted as evidence in favor of a positive association with cranial vault shape resulting in a posterior, or conditional false discovery rate (FDR) that is decreased with respect to the prior, or unconditioned FDR.”

To make the above clear in the first section of the Results, we now state:

“These results demonstrate that when a trait is genetically correlated with cranial vault shape, the information on a SNP’s positive association with that trait, increases its empirical likelihood to be positively associated with cranial vault shape.”

and further:

“Altogether, the results presented in this section clearly demonstrate that the genetic architectures of facial shape, brain shape, and BMD are enriched for associations with cranial vault shape and consequently, that GWAS data on those traits can provide additional evidence for SNPs associated with cranial vault shape.”

To clarify our conditional GWAS approach further, we added the following information to the first paragraph of the results section on “Pleiotropy-informed conditional GWAS of cranial vault shape boosts genetic discovery”:

“Analogous to the previous section, conditioning in this context refers to leveraging auxiliary GWAS data as evidence in an empirical Bayesian framework where the conditional test statistic can be formally defined as the conditional probability (i.e., the Bayesian posterior probability) that a SNP is null for cranial vault shape given that its P-values for cranial vault shape and the auxiliary phenotype are equal to or smaller than the observed P-values (Methods). This probability can be interpreted as the empirical Bayesian generalization of the FDR⁷⁴.”

Lastly, we refer to alternative methods in the discussion and explain our motivation to choose the conditional FDR method:

“...multi-trait GWAS strategies already enhance genetic insights in the meantime^{54,129–131}. In this work, we specifically opted for the conditional FDR method as it could seamlessly deal with the multivariate GWAS statistics of cranial vault shape and the other morphological phenotypes.”

Minor comments:

1. The authors have helpfully pointed out the comparisons where a 1% FDR version of the original GWAS is used. I believe this should be the main basis for comparison, not the 21 loci initially reported using the 5E-8 threshold. For simplicity and ease of understanding, I suggest removing comparisons to the original list of loci throughout.

We thank the reviewer for raising this concern and have made several changes to the text.

The main analyses that compare the results from the unconditioned and conditioned analyses, shown in Fig3b–d already use the 1% FDR version. To make this clearer, we added the following to the figure caption:

“Lead SNPs for the “unconditioned” GWAS in (b–d) were called at 1% FDR to provide a fair comparison.”

Furthermore, we dropped the comparison with the P<5e-8 version in the results section:

“In total, we identified 120 genomic loci associated with cranial vault shape at a 1% cFDR in at least one of the conditional GWAS analyses (Fig 2a, Supplementary Data 1), a marked increase compared to the 21 loci initially reported in the unconditioned GWAS. Even when re-evaluating the number of loci identified at a 1% FDR in the original, unconditioned GWAS (instead of at a P-value threshold of $< 5e-8$), only 46 loci were significant.”

into

“In total, we identified 120 genomic loci associated with cranial vault shape at a 1% cFDR in at least one of the conditional GWAS analyses (Fig 2a, Supplementary Data 1), a marked increase compared to the 46 loci identified at 1% FDR in the unconditioned GWAS.”

Additionally, we now clarify how many additional loci were identified due leveraging the auxiliary GWAS data, and how many were identified simply by using a 1% FDR threshold instead of $P < 5e-8$:

“In total, 90 of the 120 identified loci were not previously mentioned in GWAS of cranial vault morphology. Of these, 75 discoveries can be attributed to the conditional approach, while 15 were simply due to setting the threshold at 1% cFDR instead of $P < 5e-8$ ”

2. For the enrichment analysis in Figure 4, a 5% FDR version seem to have been used instead of 1% FDR. While the authors mention ‘additional loci’, I am not sure whether the enrichment analysis is performed on the loci which are present in the 5% CFDR analysis but not in the 5% FDR version of the unconditional GWAS? I think that’ll be the most appropriate set of loci to check enrichment on.

We thank the reviewer for their comment, which made us realize that our rationale could be more clearly stated in this section. We have now updated the text in accordance:

“The use of FDR-based statistics in GWAS allows for straightforward control of the number of falsely identified loci. While it is common practice in GWAS to use a very strict significance threshold to declare positive associations, we reasoned that a more relaxed threshold of 5% cFDR could help with inferring relevant biology as the additional loci should be enriched for biologically meaningful information⁷⁷. Therefore, to further explore the SNPs and genes underlying cranial vault shape, we lowered the significance threshold to 5% cFDR in each conditional GWAS and subsequently merged their genomic loci into 328 independent loci (Supplementary Data 1), explaining 8.53% of overall cranial vault shape. Following an enrichment analysis using GREAT⁷⁵, we indeed observed that most of the top 20 GO biological processes and the top 20 craniofacial mouse phenotypes identified for the initial set of 120 loci were consistent and more strongly supported among the broader set of 328 loci (Fig 4a–b; Supplementary Data 2–5). Furthermore, among the loci identified at a 5% cFDR, we identified additional genes linked to craniosynostosis in humans, including PPP3CA⁷⁸, CDKN1C⁷⁹, FOXP1⁸⁰, PRRX1⁸⁰, and ZBTB20⁸⁰. Taken together, these results show that relaxing the significance threshold to 5% cFDR helps strengthen the support for key biological processes and

pathways through the identification of additional, biologically meaningful candidate genes.”

To maximize statistical power and to provide strong evidence, it is key to run the enrichment analysis on the full set of genomic loci. Originally, this would have just been the set of 120 loci discovered at 1% cFDR, however, as reflected in the updated text, using the set of 328 loci discovered at 5% cFDR results in stronger evidence that the identified processes are related to cranial vault shape.

3. Among the new additions to the manuscript, I find this paragraph difficult to mechanistically understand.

“Still, we noted a few examples where the choice of conditional trait was directly related to the findings. Only when conditioning on BMD, we identified a locus near *LRP5* at 1% FDR, for which loss-of-function mutations are associated with osteoporosis and gain-of-function mutations with higher bone mass and craniosynostosis. Additionally, a locus near *MSX1* was only identified at 1% FDR upon conditioning on facial shape. This gene plays a critical role in facial development, including the frontal bone, with mutations linked to multiple craniofacial anomalies such as cleft lip and palate.”

If there was a gene associated with bone mass in the unconditioned analysis, which goes away when conditioning on BMD, I'd understand that the initial association was demonstrating an indirect effect of the gene on bone mass which goes away when conditioning on a trait capturing bone mass. But here the opposite seems to happen. What might be the reason for this?

We thank the reviewer for this remark and agree that this would likely be true if the conditional approach used would be based on statistical adjustment, i.e., removing covariance with the conditioning variable. However, since we use an empirical Bayesian framework, the conditioning variable is not used to reduce variance but is directly leveraged as prior knowledge or evidence. For a more detailed explanation, we kindly refer the reviewer to our response on their major comment.

As a basis for this, it can be shown for two genetically correlated traits, that a SNP's positive association with trait 1, makes it empirically more likely to be positively associated (i.e., having a lower FDR) with trait 2. In fact, we demonstrate this in Fig1c to guide the reader in building this intuition, as it is precisely this mechanism that lowered an FDR that was already close to the 1% FDR threshold to <1% cFDR at the *LRP5* locus.

We have now also emphasized this better in the results when writing about Fig1: *“These results demonstrate that when a trait is genetically correlated with cranial vault shape, the information on a SNP's positive association with that trait, increases its empirical likelihood to be positively associated with cranial vault shape.”*

4. The abstract and introduction should directly mention which traits have been conditioned on. Currently indirect phrasing is used, e.g. “these auxiliary traits were then leveraged”.

Following the reviewer's suggestions, we rephrased the introduction as:

"We then apply these principles through means of the conditional FDR method⁵⁴⁻⁶¹ to a recent cranial vault shape GWAS⁶ to leverage GWAS data on brain shape, facial shape, and BMD, thereby revealing novel associated genes and pathways. "

and the abstract as:

"Here, we use the conditional FDR method to leverage GWAS data of facial shape, brain shape, and bone mineral density to enhance SNP discovery for cranial vault shape."

5. The authors now clarify that only the European sub-sample of the ABCD data was used for analysis. This needs to be clearly stated in the Methods under 'GWAS summary data'.

The reviewer is right. We clarified this in the text:

"This includes the cranial vault GWAS in the ABCD cohort, for which ~80% of the alleles were derived from recent European ancestors⁶, which forms the basis for the main analyses performed in this work. Additionally, a version of this GWAS, conducted in a sub-sample with inferred European ancestry, was used only for LDSC. All GWAS data used in this work are freely available online. An overview of studies and sample sizes is given in Supplementary Table 1."

6. While the authors have provided some further details on the Conditional FDR method, some further details on the genomic Spearman correlations method is needed as well.

To provide additional clarity regarding the method, we added the specific Matlab function and mathematical formulas used:

"The genomic Spearman correlation (Matlab 2023a, corr with parameter 'Spearman') between both traits and using n LD blocks was calculated based on Eq. 1, where $d_i \equiv R \left[-\log_{10}(P_{i,x}) \right] - R \left[-\log_{10}(P_{i,y}) \right]$, the difference in rank between the average $-\log_{10}(P)$ -value of traits x and y in LD block i .

$$r_g(x, y) = 1 - \frac{6 \sum_{i=1}^n d_i}{n(n^2-1)} \quad (\text{Eq. 1})"$$

7. "Unsurprisingly, genomic Spearman correlations were good at predicting cross-trait enrichments of association ($R^2 = 0.685$; $P = 8.91e-4$)." The methodology is still not clear – how does one predict cross-trait enrichments of association from a genomic Spearman correlation?

We thank the reviewer for pointing this out. The statement refers to the iteratively reweighted least squares regression model as shown in Fig1c, which could be used to predict one variable from the other. However, for clarity, we have now reformulated the statement as:

“Unsurprisingly, genomic Spearman correlations were strongly correlated with cross-trait enrichments of association ($R^2 = 0.685$; $P = 8.91e-4$; Fig 1c)”, where we now also refer to Fig1c.

8. The authors have now added a summary of the biological processes behind craniofacial development. This is helpful but doesn't mention their known genetic drivers at all. E.g. do some of the major known genes affect more than one of cranial vault, face, and brain shape?

We understand the reviewer's point and have now expanded the paragraph to introduce the drivers that underly the shared genetic basis of the cranial vault, face, and brain:

“Throughout development, many genes have roles that span multiple structures of the human head, influencing both their development and integration. By controlling neural tube development, ZIC2 and ZIC3 affect the common origins of the brain and skull^{15,16}, while SOX9 has independent roles in brain¹⁷ and facial¹⁸ development. Additionally, a plethora of key facial genes (e.g., DLX5, RUNX2, and TWIST1) and signaling pathways (e.g., BMP/TGF- β , FGF, and Wnt) also affect suture fusion and are implicated in craniosynostosis^{18,21}. Together, these extensive genetic and morphological relationships are crucial to consider when studying the genetic basis of craniofacial variation as they present both opportunities and challenges for genetic discovery and interpretation.”

Reviewer #3 (Remarks to the Author):

In replying to my major point, I'm sure it became abundantly clear to the authors that this reviewer had completely misunderstood what 'conditional GWAS' meant in their context! I presume they've been rather exasperated, although they've been extremely polite in their replies, and their current response clarifies this confusion substantially. It might be useful to acknowledge though, that many other readers would come from a background like mine where 'conditional GWAS' is known to refer to the kind of adjustment-based regression approach used in GCTA-COJO (Yang et al. Nature Genetics 2012) and also in other quantitative genetics approaches such as Conditional eQTL (Dobbyn et al. AJHG 2018). While one can see that the word 'conditional' in the context of FDR arises from conditional probabilities, converting it to 'conditional GWAS' in this manuscript seems to have started the confusion. I note that this phrase was not used in the original paper [ref 54] which proposed the cFDR procedure. Perhaps the authors can consider changing the phrase. The word 'conditional' in front of GWAS in the title is perhaps unnecessary as 'pleiotropy-informed GWAS' is clear enough. But generally they might want to find a better name for this GWAS procedure, something like 'conditional FDR-based GWAS' or 'cFDR-GWAS' throughout the manuscript should help to clear up the confusion.

We acknowledge this point fully and thank the reviewer for suggesting alternatives. We have now removed 'conditional' whenever it was followed by 'pleiotropy-informed' such as in the title and subtitles. Throughout the text, we replaced 'conditional GWAS' by 'cFDR-GWAS'.

The authors have provided additional clarifications on Minor point 2. I have one further question: the enrichment analysis seems to have been done on all loci that were significant (at 5%) in the cFDR-based GWAS. As they are talking about 'additional loci', I would expect that this enrichment analysis would be done on loci excluding those found at 5% FDR in the original GWAS? Can this be clarified please.

The reviewer is right that the analysis was done on all loci significant at 5% cFDR. We understand that 'additional loci' may be ambiguous and changed the wording to avoid confusion.

"we reasoned that a more relaxed threshold of 5% cFDR could help with inferring relevant biology as the additional loci should be enriched for biologically meaningful information⁷⁷."

was changed to:

"we reasoned that relaxing the threshold to 5% cFDR (instead of 1% cFDR) would yield a larger set of genomic loci, still enriched for biologically meaningful information⁷⁷, that could help to more robustly implicate biological processes and pathways."